# p21-activated kinase 4 suppresses fatty acid β-oxidation and ketogenesis by phosphorylating NCoR1

Min Yan Shi[1], Hwang Chan Yu[1], Chang Yeob Han[2], In Hyuk Bang[1], Ho Sung Park[3], Kyu Yun Jang [3], Sangkyu Lee[4], Jeong Bum Son [5], Nam Doo Kim[5], Byung-Hyun Park [1] ✉ & Eun Ju Bae[2] ✉

PPARα corepressor NCoR1 is a key regulator of fatty acid β-oxidation and ketogenesis. However, its regulatory mechanism is largely unknown. Here, we report that oncoprotein p21-activated kinase 4 (PAK4) is an NCoR1 kinase. Specifically, PAK4 phosphorylates NCoR1 at T1619/T2124, resulting in an increase in its nuclear localization and interaction with PPARα, thereby repressing the transcriptional activity of PPARα. We observe impaired ketogenesis and increases in PAK4 protein and NCoR1 phosphorylation levels in liver tissues of high fat diet-fed mice, NAFLD patients, and hepatocellular carcinoma patients. Forced overexpression of PAK4 in mice represses ketogenesis and thereby increases hepatic fat accumulation, whereas genetic ablation or pharmacological inhibition of PAK4 exhibites an opposite phenotype. Interestingly, PAK4 protein levels are significantly suppressed by fasting, largely through either cAMP/PKA- or Sirt1-mediated ubiquitination and proteasome degradation. In this way, our findings provide evidence for a PAK4-NCoR1/PPARα signaling pathway that regulates fatty acid β-oxidation and ketogenesis.

Ketogenesis represents an important adaptive mechanism in health and disease. Ketone bodies are primarily produced in the liver and used exclusively in the extrahepatic organs such as the brain, heart, and skeletal muscle as an alternative fuel during starvation or prolonged exercise. They also function as endogenous or extracellular signaling molecules. Ketone bodies induce epigenetic changes that alter gene expression and regulate cellular signaling metabolites, collectively serving as a central node for an organism's energy metabolism[1]. During fasting or in the absence of insulin, triglyceride (TG) stored in adipocytes is hydrolyzed into free fatty acids (FFAs) and glycerol. FFAs are taken up into the liver where they are then β-oxidized to acetyl-CoA and further metabolized to ketone bodies.

At a subcellular level, the generation of ketone bodies occurs in the mitochondria via a series of enzymatic reactions, during which 3-hydroxymethylglutaryl-CoA synthase 2 (HMGCS2) catalyzes the rate-determining conversion of acetyl-CoA into ketone bodies. Peroxisome proliferator-activated receptor alpha (PPARα) plays a central transcription role in the diverse mechanisms that coordinate the expression and enzymatic activity of HMGCS2[2]. Consequently, *Ppara* knockout mice fail to induce HMGCS2 in the liver during fasting and exhibit hypoketonemia and hepatic steatosis[3,4]. PPARα transactivation in hepatocytes varies with changes in the nutritional and hormonal milieu. Fasting hormone glucagon activates PPARα, which then induces the transcription of genes involved in fatty acid β-oxidation and ketogenesis[5]. Conversely, insulin blocks ketogenesis through PPARα

[1]Department of Biochemistry and Molecular Biology, Jeonbuk National University Medical School, Jeonju 54896, Republic of Korea. [2]School of Pharmacy, Jeonbuk National University, Jeonju 54896, Republic of Korea. [3]Department of Pathology, Jeonbuk National University Medical School, Jeonju 54896, Republic of Korea. [4]School of Pharmacy, Sungkyunkwan University, Suwon 16419, Republic of Korea. [5]VORONOI BIO Inc., Incheon 21984, Republic of Korea. ✉e-mail: bhpark@jbnu.ac.kr; ejbae7@jbnu.ac.kr

inhibition followed by activation of mechanistic target of rapamycin complex 1 (mTORC1)[6]. PPARα activity is further regulated by interactions with co-regulators. In the fed state, PPARα interacts with nuclear receptor corepressor 1 (NCoR1), which suppresses ketogenic gene expression, while fasting releases PPARα from NCoR1, thereby resuming ketogenesis[7]. These processes highlight the importance of the interplay between PPARα and NCoR1. The ability of NCoR1 to bind to PPARα is interrupted by post-translational modifications; while ubiquitination of NCoR1 facilitates its exchange for co-activators[8], phosphorylation triggers its nuclear import, with multiple kinases reportedly involved in these actions[6,7,9].

Impairments in ketogenesis are frequently observed in individuals with non-alcoholic fatty liver disease (NAFLD)[10,11]. The excess of fatty acids relative to the liver's ability to produce ketone bodies, associated with obesity, drives the metabolic pathway toward lipogenesis, thereby contributing to ectopic fat accumulation[1]. The overexpression of HMGCS2 improves high fat diet (HFD)-induced NAFLD in mice, whereas the deletion of HMGCS2 leads to the opposite phenotype[12], confirming the role of ketogenesis in preventing the development of NAFLD. Similarly, a failure to convert excess acetyl-CoA into ketone bodies increases the gluconeogenic burden on the liver and eventually leads to hyperglycemia[10]. Furthermore, while findings have been somewhat inconsistent, impaired ketogenesis has been observed in hepatocellular carcinoma (HCC) patients. The defect is accompanied by decreases in the expression levels of HMGCS2 and 3-hydroxybutyrate dehydrogenase 1 (BDH1), an enzyme catalyzing the interconversion between the two major ketone bodies acetoacetate and β-hydroxybutyrate (βOHB)[13–15]. This suggests that dysregulated ketogenesis may be linked to tumorigenesis. While ketogenesis represents an important adaptive process that regulates not only glucose and lipid metabolism but also cancer development, the molecular mechanisms underlying its dysregulation remain incompletely understood.

Serine/threonine kinase p21-activated kinase 4 (PAK4), belonging to the PAK family (PAK1-6), is best known for its regulation of cytoskeletal changes, migration, and cancer cell proliferation and invasion[16]. Its overexpression has been reported in conjunction with various types of human metastatic cancers with aggressive features[17]. Recently, we revealed new functions of PAK4 that go beyond its oncogenic role. For example, PAK4 impairs skeletal muscle regeneration by phosphorylating PPARγ and enhancing the expression of phosphatase and tensin homolog[18]. In hepatocytes, PAK4 phosphorylates nuclear factor erythroid 2-related factor 2, which leads to its nuclear export and protein destabilization, thus aggravating ischemia-reperfusion injury[19]. However, little is known about the function of PAK4 in energy homeostasis. Based on previous observations that cancer cells often overexpress PAK4 and exhibit altered metabolic flexibility, we hypothesized that PAK4 may regulate metabolic adaptation processes. To focus our investigation on the regulatory potential of PAK4 for fatty acid oxidation and ketogenesis, we generated hepatocyte-specific *Pak4* knockout (*Pak4[flox/flox];Albumin-Cre*) mice and subjected them to fasting- and high-fat-very-low-carbohydrate ketogenic diet (KD) feeding-ketosis models. Additionally, we employed a small molecule PAK4 inhibitor ND201651[19] as a model compound to establish a proof of concept. We also tested our hypotheses in humans and found that PAK4 protein levels were upregulated in the liver tissue of NAFLD and HCC patients. Specifically, we found that in the tumor tissues of HCC patients, PAK4 protein expression inversely correlated with ketone body levels.

## Results

### PAK4 protein levels are suppressed by fasting through distinct pathways involving PKA and Sirt1

To gain insight into the role of PAK4 in the control of metabolic flexibility in hepatocytes, we first analyzed PAK4 expression levels in liver tissues under various conditions of energy stress. Fasting for 6–36 h gradually suppressed PAK4 protein levels but refeeding restored them (Fig. 1a). Interestingly, the changes in *Pak4* mRNA levels did not correspond with the fluctuations observed in the protein levels (Supplementary Fig. 1a). PAK4 protein levels revealed a reciprocal pattern with blood levels of glucagon and ketone body β-hydroxybutyrate (βOHB) (Supplementary Fig. 1b). Mice on a KD also exhibited significantly lower PAK4 protein, but not *Pak4* mRNA levels, in the liver than those fed a normal chow diet (NCD) (Fig. 1b and Supplementary Fig. 1c). Conversely, mice with fatty liver conditions, such as those fed a HFD, *ob/ob* mice, and *db/db* mice, exhibited elevated levels of PAK4 protein in their liver tissues compared to the control mice (Fig. 1c and Supplementary Fig. 1d).

We then tested the direct effects of fasting- and fed-hormones on PAK4 protein levels and found that glucagon and epinephrine consistently repressed PAK4 protein levels in primary hepatocytes, whereas insulin had no effect (Supplementary Fig. 1e). Notably, the suppression of PAK4 protein by glucagon was rescued by the PKA inhibitor H89 or insulin, which was associated with Ser/Thr phosphorylation of PAK4 (Supplementary Fig. 1f). We then hypothesized that glucagon represses PAK4 protein levels through proteasome-dependent degradation, as we observed a reversal of PAK4 repression upon treatment with MG132, a proteasome inhibitor (Supplementary Fig. 1g). In line with this, treatment with glucagon increased the level of ubiquitinated PAK4 (Fig. 1d) and resulted in decreased protein stability of PAK4 through a PKA-dependent manner (Fig. 1e).

Since KD feeding repressed PAK4 protein levels (Fig. 1b), we proceeded to investigate the regulation of PAK4 protein by ketone bodies. Incubation of hepatocytes with immediate substrate for ketogenesis, octanoate[1], suppressed PAK4 protein levels in an HMGCS2-dependent manner (Supplementary Fig. 1h). The absence of changes in *Pak4* mRNA levels with octanoate suggests that the regulation of PAK4 is post-transcriptional level (Supplementary Fig. 1i). βOHB has been shown to regulate various cellular functions as an endogenous modulator of histone deacetylase (HDAC), inhibiting class I/II/IV HDACs[20] or activating class III HDAC Sirt1[21]. We tested the involvement of HDAC in regulating PAK4 protein stability. Results showed that the repression of PAK4 induced by octanoate was abolished by panobinostat (a pan-inhibitor of the HDAC family)[22], but not by pracinostat (a class I/II/IV HDAC inhibitor) (Supplementary Fig. 1j), indicating a role for class III HDAC sirtuins in PAK4 downregulation. Octanoate specifically and significantly upregulated the expression of Sirt1, as previously reported[21], with no effect on other sirtuins in hepatocytes (Supplementary Fig. 1k). Treatment with the long-chain fatty acid palmitate or βOHB also resulted in a reduction in PAK4 protein levels, showing an inverse correlation with Sirt1 protein levels (Supplementary Fig. 1l). Palmitate treatment confirmed an increase in ketone body synthesis, although to a lesser degree compared to octanoate (Supplementary Fig. 1m). Importantly, the decrease in PAK4 protein levels induced by octanoate was reversed upon Sirt1 silencing (Supplementary Fig. 1n). Similar to the findings in the glucagon study, treatment with octanoate also led to PAK4 protein degradation via the ubiquitination pathway, and this process was dependent on Sirt1 activation (Fig. 1f, g).

Considering the apparent link between the cAMP-PKA pathway and Sirt1 activation[23,24], we further investigated the regulatory connections between these two pathways in PAK4 degradation. Glucagon treatment induced the Ser/Thr phosphorylation and activation of Sirt1 (deacetylation of FoxO1), which were also counteracted by either H89 or insulin (Supplementary Fig. 1o), implying the involvement of the cAMP-PKA-Sirt1 axis in PAK4 degradation. However, treatment with the Sirt1 inhibitor EX-527 did not impact the degradation of PAK4 by glucagon (Supplementary Fig. 1p), thus ruling out the possibility of Sirt1 activation being responsible for glucagon-mediated PAK4 repression.

To determine the direct causality of Ser/Thr phosphorylation on PAK4 in protein degradation, we mutated candidate serine sites

with alanine. The results indicated that S258 of PAK4 is the critical site responsible for PKA activation-induced ubiquitination and degradation of PAK4 (Fig. 1h, i). To further identify the specific sites of ubiquitination in PAK4, we substituted four known lysine residues with alanine. The results revealed that K31, K540, and K546 of PAK4 are involved in the ubiquitination and subsequent proteasomal degradation (Supplementary Fig. 2a, b). As a previous study has demonstrated that PAK4 interacts with murine double minute 2 (MDM2), an oncogenic E3 ubiquitin ligase that promotes p53 tumor suppressor ubiquitination[25], we further hypothesized that glucagon-mediated ubiquitination of PAK4 would be associated with MDM2. Inhibition of MDM2 with nutlin-3 or I3C, but not treatment with heclin (a HECT ligase inhibitor), reversed the inhibitory effect of glucagon on PAK4 protein (Supplementary Fig. 2c). The results from an additional study using siRNAs against MDM2 or NEDD4, the latter of which is also reported to cooperate with PAK4[26], demonstrated that MDM2 functions as a E3 ubiquitin ligase to target PAK4 for degradation (Fig. 1j and

Supplementary Fig. 2d). These results suggest that glucagon-mediated cAMP-PKA pathway and βOHB-mediated Sirt1 upregulation independently play roles in the ubiquitination/proteasome-mediated degradation of PAK4 (Fig. 1k).

## Adenoviral overexpression of PAK4 suppresses fasting- and KD-induced ketogenesis

To understand how PAK4 downregulation during ketogenic states contributes to adaptive responses to fasting, we introduced either wild-type PAK4 or kinase-inactive mutant PAK4^(S474A) into mouse livers via adenoviral vectors. The overexpression of PAK4, but not PAK4^(S474A), significantly reduced fasting βOHB blood levels compared to the AdLacZ group, with no changes observed in body weight or blood glucose levels (Fig. 2a and Supplementary Fig. 3a, b). Furthermore, serum and hepatic triglyceride (TG) levels, gross liver morphology, and histochemical analysis (H&E and Oil Red O staining) demonstrated that liver steatosis induced by fasting or KD-feeding was exacerbated by

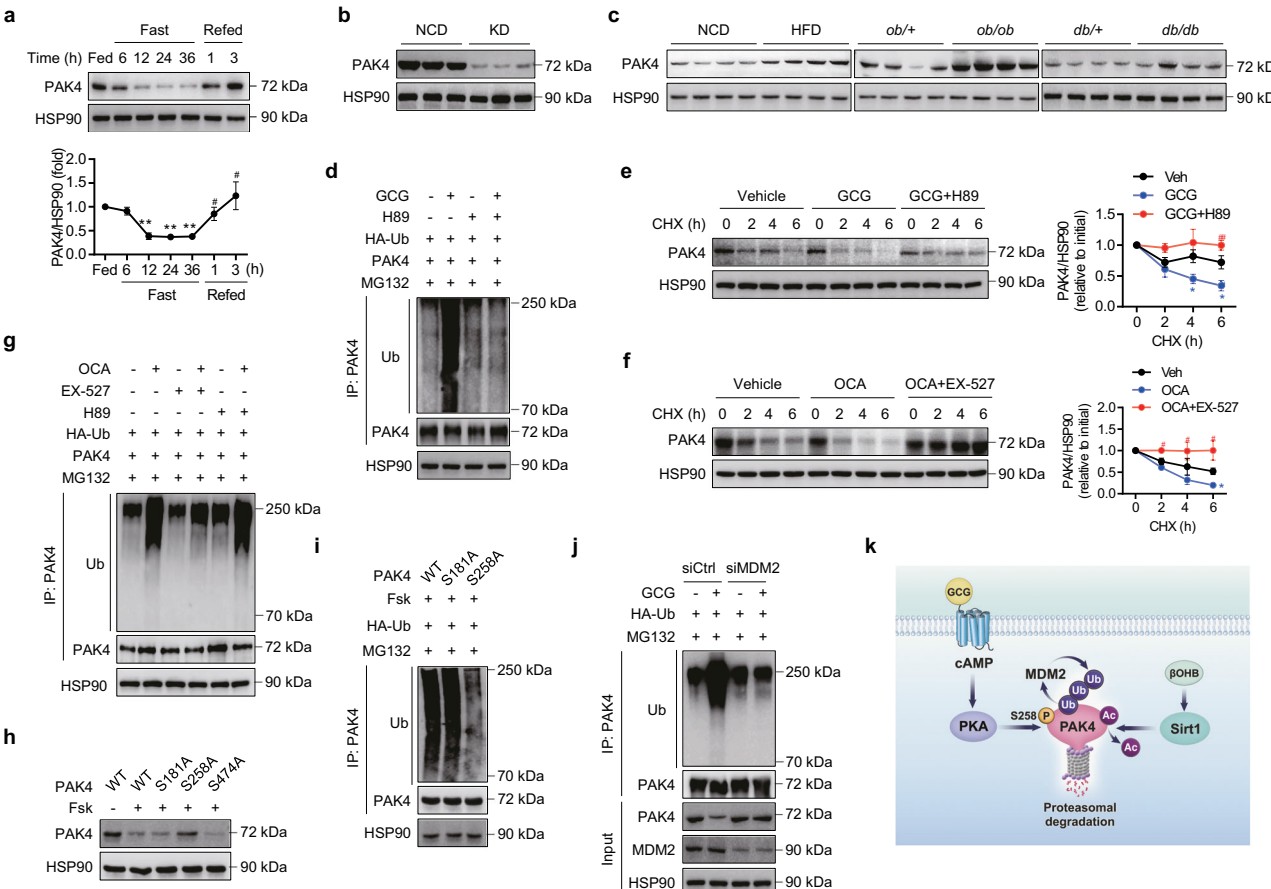

**Fig. 1 | PAK4 protein levels are repressed by fasting. a** Eight-week-old male C57BL/6 mice were either fed a normal chow diet (NCD) *ad libitum*, fasted for 6–36 h, or refed for 1 or 3 h following a 36 h fast. Liver tissues were subjected to Western blotting to determine PAK4 protein levels ($n = 3$). **P < 0.01 versus fed; #P < 0.05 versus fasted for 36 h. **b** Eight-week-old male C57BL/6 mice were fed either a NCD or a ketogenic diet (KD) for 2 weeks, and hepatic PAK4 protein levels were analyzed. **c** Representative immunoblot images for PAK4 expression in liver tissue obtained from mice fed either a NCD or a 60% high-fat diet (HFD) for 16 weeks or from *ob/ob* or *db/db* mice and their control groups. **d** AML12 cells transfected with HA-Ub and PAK4 were treated with glucagon (GCG, 100 nM) for 12 h in the presence or absence of H89 (10 μM). Cell lysates were immunoprecipitated with anti-PAK4 antibody and immunoblotted with anti-ubiquitin (Ub) antibody. **e, f** Mouse primary hepatocytes were treated with glucagon (100 nM) with or without H89 (10 μM), or octanoate (OCA, 2 mM) with or without EX-527 (100 nM). Subsequently, the cells were treated with cycloheximide (CHX, 100 μg/ml) for the indicated time periods.

The protein levels of PAK4 were compared ($n = 3$). *P < 0.05 versus vehicle (Veh); #P < 0.05 and ##P < 0.01 versus GCG or OCA. **g** Primary hepatocytes were treated with octanoate (2 mM) for 12 h in the presence or absence of H89 (10 μM) or EX-527 (100 nM). Cell lysates were immunoprecipitated with anti-PAK4 antibody and immunoblotted with anti-ubiquitin (Ub) antibody. **h, i** AML12 cells were transfected with wild-type (WT) or mutant PAK4 (S181A, S258A, or S474A) and then treated with forskolin (Fsk, 10 μM) for 12 h to compare protein degradation and ubiquitination of PAK4. **j** AML12 cells transfected with either control siRNA or siRNA targeting MDM2 were treated with glucagon (100 nM) for 12 h. Cell lysates were immunoblotted for indicated proteins or immunoprecipitated with anti-PAK4 antibody followed by immunoblotting with anti-ubiquitin (Ub) antibody. **k** Schematic summary of PKA- and Sirt1-mediated PAK4 degradation. Data are presented as the mean ± SEM. One-way ANOVA followed by Dunnett's multiple comparisons test (**a**) and Tukey's multiple comparisons test (**e**, **f**) were conducted for statistical analyses. Source data are provided as a Source Data file.

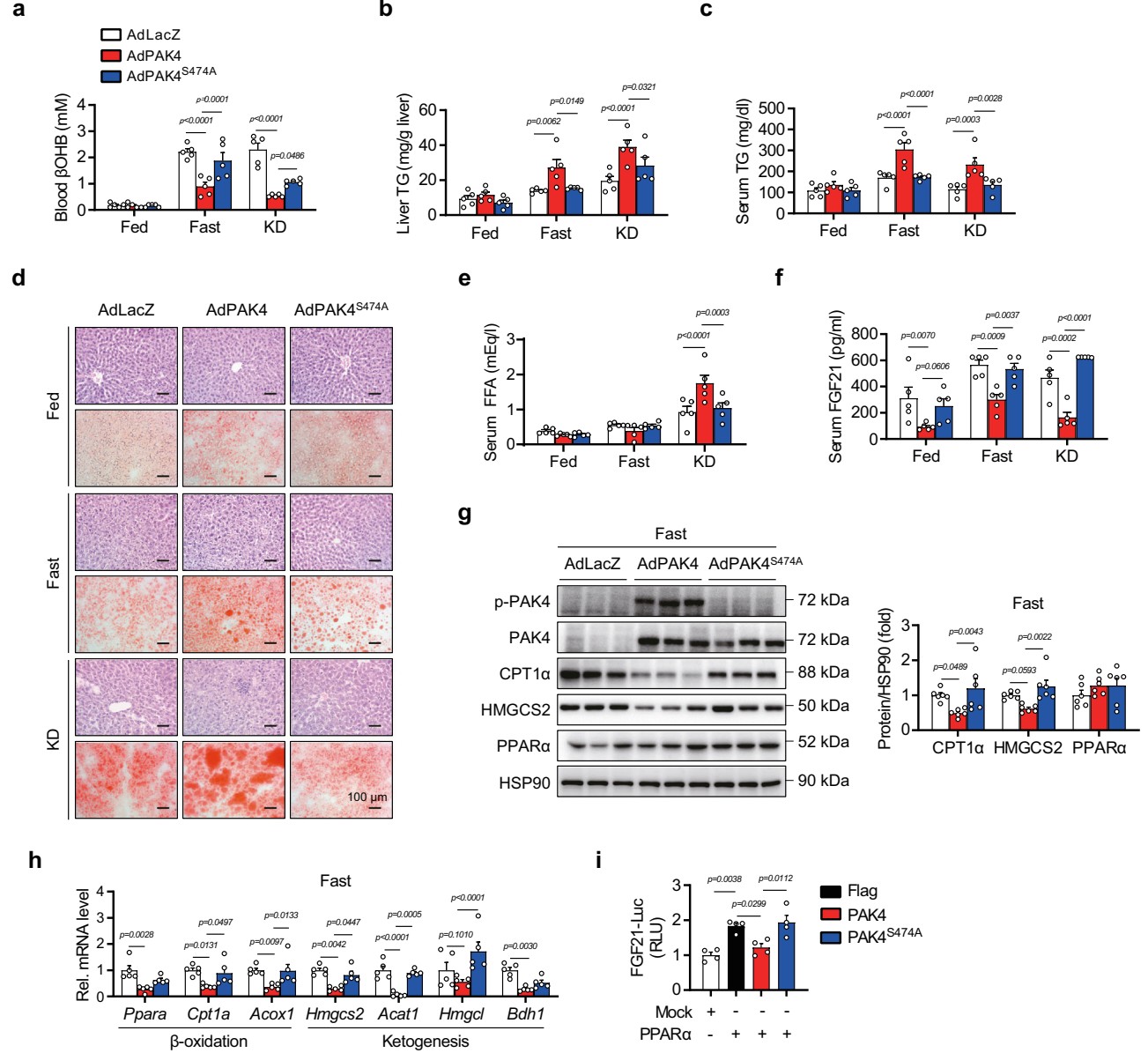

**Fig. 2 | Forced overexpression of PAK4 in mouse liver attenuates ketogenic responses.** Eight-week-old male C57BL/6 mice were injected with adenoviruses expressing control (AdLacZ), PAK4 (AdPAK4), or kinase-inactive mutant PAK4$^{S474A}$ (AdPAK4$^{S474A}$) and then fed a normal chow diet (NCD) *ad libitum* (Fed), fasted for 24 h (Fast), or fed a ketogenic diet (KD) for 2 weeks. **a–c** Blood levels of βOHB (**a**) and TG levels in the liver (**b**) and serum (**c**) were compared ($n = 5$). **d** Representative microscopic images of hematoxylin & eosin (H&E, top)- or Oil red O (ORO, bottom)-stained liver tissue (scale bars, 100 μm). **e, f** Serum levels of free fatty acid (**e**, $n = 5$)

and FGF21 (**f**, $n = 5$). **g, h** Western blotting (**g**, $n = 6$) and qPCR (**h**, $n = 5$) analyses of liver tissue obtained from fasted mice. **i** AML12 cells were co-transfected with PPARα expression plasmid, FGF21 promoter luciferase reporter plasmid, and the plasmids encoding PAK4 or PAK4$^{S474A}$. FGF21-luciferase activities were measured and expressed as the fold change relative to Mock ($n = 4$). RLU relative luminescence unit. Data are presented as the mean ± SEM. One-way ANOVA followed by Tukey's multiple comparisons test was conducted for statistical analyses (**a–c**, **e–i**). Source data are provided as a Source Data file.

PAK4 overexpression, but not PAK4$^{S474A}$ (Fig. 2b–d and Supplementary Fig. 3c). Serum levels of FFA were increased by PAK4 overexpression only under KD-feeding (Fig. 2e). PAK4 overexpression repressed serum levels of fibroblast growth factor 21 (FGF21), a critical endocrine hormone for ketogenesis[27] (Fig. 2f), suggesting a possible role for FGF21 in mediating PAK4's inhibition of ketogenesis. PAK4 overexpression also led to defects in fatty acid β-oxidation and ketogenesis, as evidenced by reduced levels of carnitine palmitoyltransferase 1A (CPT1α) and HMGCS2 protein expression (Fig. 2g). Additionally, mRNA levels of genes responsible for fatty acid β-oxidation (*Ppara*, *Cpt1a*, and *Acox1*) and ketogenesis (*Hmgcs2*, *Acat1*, and *Bdh1*) were downregulated by PAK4 overexpression with no changes in the level of genes of de novo lipogenesis (Fig. 2h and Supplementary Fig. 3d). The same inhibitory

effect of PAK4 on ketogenesis and ketogenic gene expression was also observed in in vitro hepatocyte culture models (Supplementary Fig. 3e, f). Consistent with the results of serum level of FGF21 in mice, the overexpression of PAK4 but not of PAK4$^{S474A}$ attenuated FGF21-luciferase activity (Fig. 2i). Taken together, these findings suggest that PAK4 overexpression in the liver impairs fasting- and KD-induced ketogenic responses, and this effect is dependent on PAK4 kinase activity.

## Hepatic deficiency of PAK4 enhances ketogenesis
To investigate how PAK4 controls ketogenesis, we conducted an integrated RNA-Seq analysis using the livers of wild-type (WT) and *Pak4* liver-specific knockout (LKO) male mice in a fasted state.

Interestingly, the gene ontology analysis of the differentially expressed genes (DEGs) revealed that the top 20 enriched terms were related to lipid, fatty acid, and monocarboxylic acid metabolism (Supplementary Fig. 4a, b). Specifically, we found that the upregulated DEGs were enriched in terms related to lipid catabolic and ketone metabolic processes, while the downregulated DEGs were enriched in terms associated with lipid biosynthetic processes and known functions of PAK4 such as cell cycle and cytoskeleton organization (Supplementary Fig. 4c). In addition, the gene set enrichment analysis (GSEA) and heatmap analysis showed that several gene sets involved in regulating ketone biosynthesis, oxidative phosphorylation, and cellular responses to starvation were upregulated in the liver of *Pak4* LKO mice compared to WT mice (Fig. 3a and Supplementary Fig. 4d).

To elaborate upon this finding, we next performed in vivo luciferase bioluminescence imaging to measure *Hmgcs2* promoter activity after injecting AdHmgcs2-PPRE-Luc into the mouse liver. Fasting for 24 h or KD-feeding for three days increased *Hmgcs2* promoter activity by 6.5-fold and 14.4-fold, respectively, compared to that observed in NCD-fed WT mice (Fig. 3b). These changes were even greater in LKO mice compared to WT control animals (13.7- and 27.0-fold increases in fasted and KD mice, respectively), consistent with earlier *Pak4* overexpression studies. We further compared the metabolic phenotypes of WT and LKO mice. Circulating βOHB levels under NCD-fed, fasted, and KD-fed conditions were increased in LKO mice compared to WT mice (Fig. 3c). Body weight as well as liver, epididymal adipose tissue (EAT), and gastrocnemius muscle wet weight remained unaltered across genotypes (Supplementary Fig 4e–h). In line with the increases in blood βOHB levels, hepatic and serum TG levels were significantly suppressed in *Pak4* LKO mice in response to fasting and KD feeding compared to WT control animals (Fig. 3d, e and Supplementary Fig. 4i, j). In harmony with these findings, hepatic stress was reduced in *Pak4* LKO mice under KD conditions, as evidenced by the mRNA levels of inflammation-related genes and protein levels of endoplasmic reticulum stress genes (Supplementary Fig. 4k, l). Serum levels of FFA were lower in LKO mice under fasting or KD-feeding conditions compared to those in WT mice, whereas FGF21 was significantly higher in fed-LKO mice compared to fed-WT mice (Fig. 3f, g). Protein levels of CPT1α, HMGCS2, and PPARα, as well as mRNA levels of β-oxidation- and ketogenesis-related genes, were significantly increased in liver tissues of fasted *Pak4* LKO mice compared to fasted-WT mice (Fig. 3h, i). However, no significant change in the expression of de novo lipogenesis genes was observed between the genotypes (Supplementary Fig. 4m). Similar changes were observed after KD feeding (Fig. 3j, k). Lastly, when primary hepatocytes from WT and LKO mice were treated with octanoate, βOHB production was increased by PAK4 ablation (Supplementary Fig. 4n). Most of the phenotypes observed in male *Pak4* LKO mice under fasting or KD feeding conditions were also observed in female *Pak4* LKO mice (Supplementary Fig. 5a–h). These results indicate that the loss of hepatic PAK4 results in an enhancement of β-oxidation and ketogenesis, which contributes to the attenuation of fatty liver.

## PAK4 suppresses PPARα activity through NCoR1 phosphorylation

KEGG pathway analysis further demonstrated that the PPAR signaling pathway, in conjunction with the fatty acid metabolism, was significantly altered in *Pak4* LKO mice (Fig. 4a). We identified PPARα as a key molecule that interacts with genes involved in the signaling pathway (Fig. 4b). Consequently, our focus shifted to investigating how PAK4 regulates PPARα.

In AML12 cells co-transfected with a peroxisome-proliferator-responsive element (PPRE)×3-Luc reporter plasmid and a PPARα expression vector, PAK4, but not PAK4$^{S474A}$, suppressed PPARα luciferase activity (Fig. 4c). The transcriptional activity of PPARα is mainly

regulated through interactions with co-activators or co-repressors[28]. We ruled out the possibility that PAK4 directly phosphorylates PPARα since phosphorylation of PPARα either leads to its degradation or correlates with increased transactivation[29]. Proximity ligation assays (PLA) and co-IP studies revealed that PAK4 facilitated the interaction between PPARα and the corepressor NCoR1 (Fig. 4d, e). On the contrary, the binding of NCoR1 to the other nuclear receptor, thyroid hormone receptor beta (THRβ), was decreased by PAK4 over-expression, showing a reciprocal regulation compared to PPARα. The interactions between NCoR1 with LXRα, PPARα with p300, or PPARα with SMRT remained unchanged (Fig. 4d, e). Considering that the NCoR1-THRβ pathway is crucial for hepatic lipid synthesis and storage[30,31], further study is needed to understand the regulation of PAK4 on this pathway. Confocal microscopic and Western blot analyses showed that PAK4 overexpression increased nuclear levels of NCoR1 while reducing cytosolic levels (Fig. 4f, g). Using ChIP-qPCR, we determined NCoR1 recruitment to the PPRE promoter regions of *Ppara*, *Cpt1a*, and *Hmgcs2* and found that PAK4 promoted the binding of NCoR1 to these sequences (Fig. 4h). To test the causal relationship between PAK4 regulation of PPARα activity and NCoR1 function, we conducted NCoR1 knockdown experiments. In accordance with previous reports on liver-specific *Ncor1* KO mice[7] and silencing of NCoR1 in HepG2 cells[32], silencing NCoR1 led to a significant induction of PPRE-luciferase activity (Fig. 4i, j). More importantly, the repression of PPRE promoter activity and the expression of *Cpt1a*, *Acox1*, and *Hmgcs2* by PAK4 was abolished in cells silenced with NCoR1 (Fig. 4k). Together, these observations suggest that PAK4 inhibits PPARα activity by promoting NCoR1 recruitment to PPARα.

We were thus interested in the possibility that PAK4 phosphorylates NCoR1. NCoR1 phosphorylation detected by anti-p-Ser/Thr universal antibody in NCoR1 immunoprecipitates was drastically increased by PAK4 overexpression, and an in vitro kinase assay confirmed that PAK4 indeed directly phosphorylated NCoR1 (Fig. 5a, b). PhosphoNET prediction identified three residues (S837, T1619, and T2124) of mouse NCoR1 that can be phosphorylated by PAK4, and moreover, these PAK4 sites were conserved among vertebrate NCoR1 sequences queried (Fig. 5c). We therefore performed a competition assay employing all peptides that contained one of these three residues and found that NCoR1 phosphorylation by PAK4 was reduced in the presence of any of these competitors (Fig. 5c). We thus manufactured phospho-inactive mutants of NCoR1 to evaluate the role of NCoR1 phosphorylation in PAK4's repression of PPARα activity. PAK4 did not exhibit any effect on PPRE-luciferase activity upon over-expression of either T1619A, T2124A, or T1619A/T2124A NCoR1 mutants (Fig. 5d). However, PAK4's repression of PPRE-luciferase activity was reserved by overexpression of S837A mutant. Phosphorylation of NCoR1 on T1619 and T2124 were further confirmed by LC-MS/MS analysis (Supplementary Fig. 6a). Altogether, this suggests that PAK4's phosphorylation of NCoR1 on T1619/T2124 residues leads to its recruitment to PPARα.

To further validate the functional significance of NCoR1 phosphorylation at T1619/T2124, we conducted in vitro and in vivo investigations using its phospho-inactive mutant. Through a co-IP study following transfection with either WT or a double-mutant (T1619A/T2124A) form of NCoR1, we observed that PAK4-mediated phosphorylation of NCoR1, its physical interaction with PPARα, and its accumulation within the nucleus were attenuated in cells transfected with the T1619A/T2124A mutant (Fig. 5e). Interestingly, we also noted a reduction in the total level of NCoR1 in cells transfected with the double-mutant. In addition, NCoR1 ubiquitination was dramatically increased in cells overexpressing double-mutant (Fig. 5f). In mouse livers, we observed similar trends as observed in cells, where NCoR1 ubiquitination slightly increased during fasting but markedly increased in LKO mice (Fig. 5g). As a result, under fasting conditions, the phosphorylation and nuclear localization of NCoR1 were suppressed compared to the fed

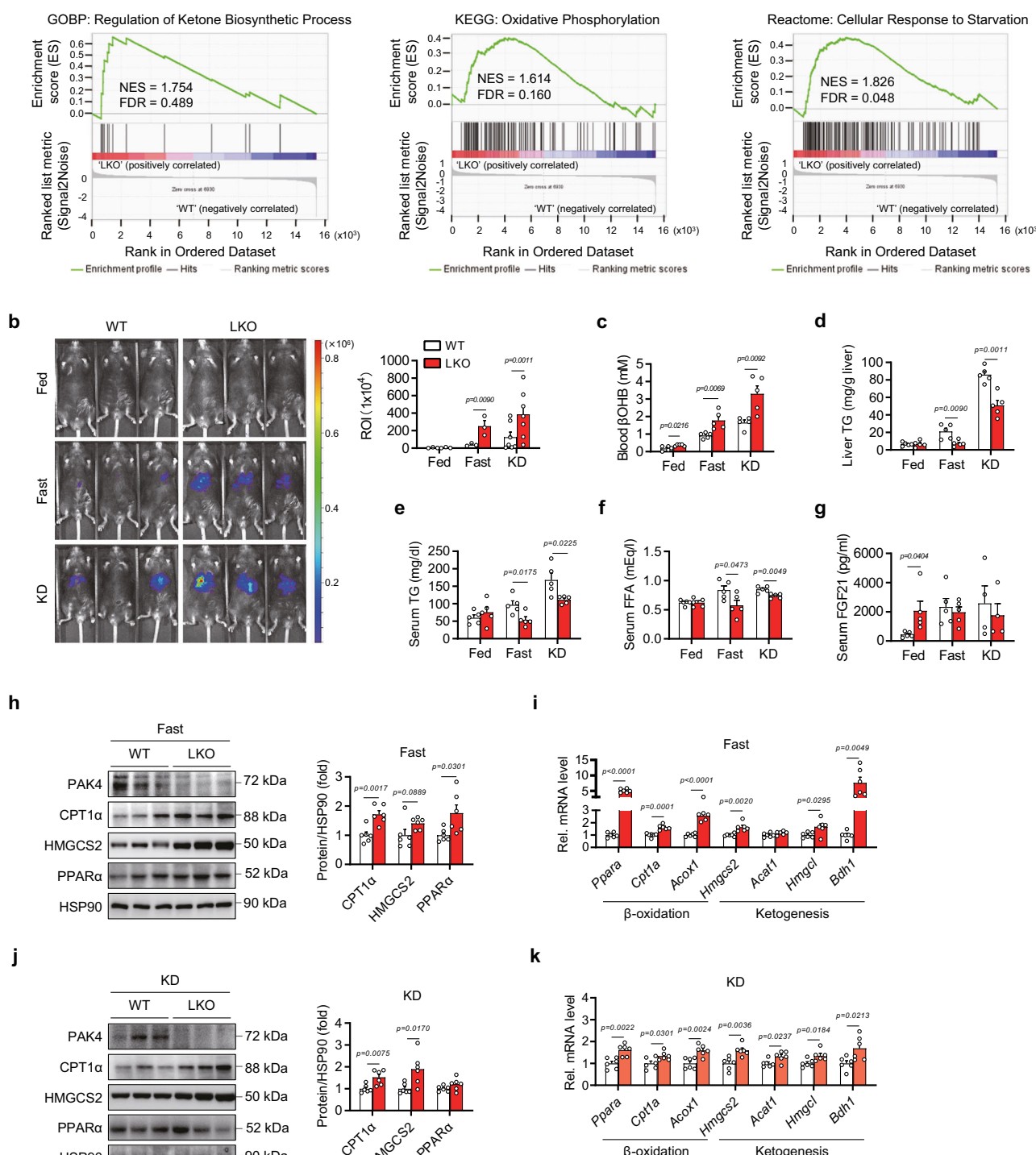

**Fig. 3 | Hepatocyte-specific PAK4 deficiency enhances ketogenic responses.**
**a** Gene set enrichment analysis plots of RNA-Seq data. Eight-week-old male *Pak4* LKO and WT mice were fasted for 24 h, and RNA samples from liver tissues were subjected to RNA-Seq. **b** Representative IVIS images of in vivo luciferase activity after injection with adenovirus expressing HMGCS2-luciferase in *Pak4* LKO and WT mice (*n* = 3 for Fed and Fast groups and *n* = 7 for KD group). **c**–**g** *Pak4* LKO and WT mice were fed a normal chow ad libitum (Fed), fasted for 24 h (Fast), or fed a ketogenic diet (KD) for 2 weeks. Blood levels of βOHB (**c**, *n* = 5), TG levels in the liver

(**d**, *n* = 5) and serum (**e**, *n* = 5), and serum levels of free fatty acid (**f**, *n* = 5) and FGF21 (**g**, *n* = 5 for Fed and Fast, *n* = 4 for KD) were compared. **h**–**k** Western blotting analysis of liver tissue obtained from *Pak4* LKO and WT mice after fasting (**h**, *n* = 6) or KD-feeding (**j**, *n* = 6), and qPCR analysis of liver tissue after fasting (**i**, *n* = 6) or KD-feeding (**k**, *n* = 6). Data are presented as the mean ± SEM. Unpaired two-tailed *t* test was conducted for statistical analyses (**b**–**k**). Source data are provided as a Source Data file.

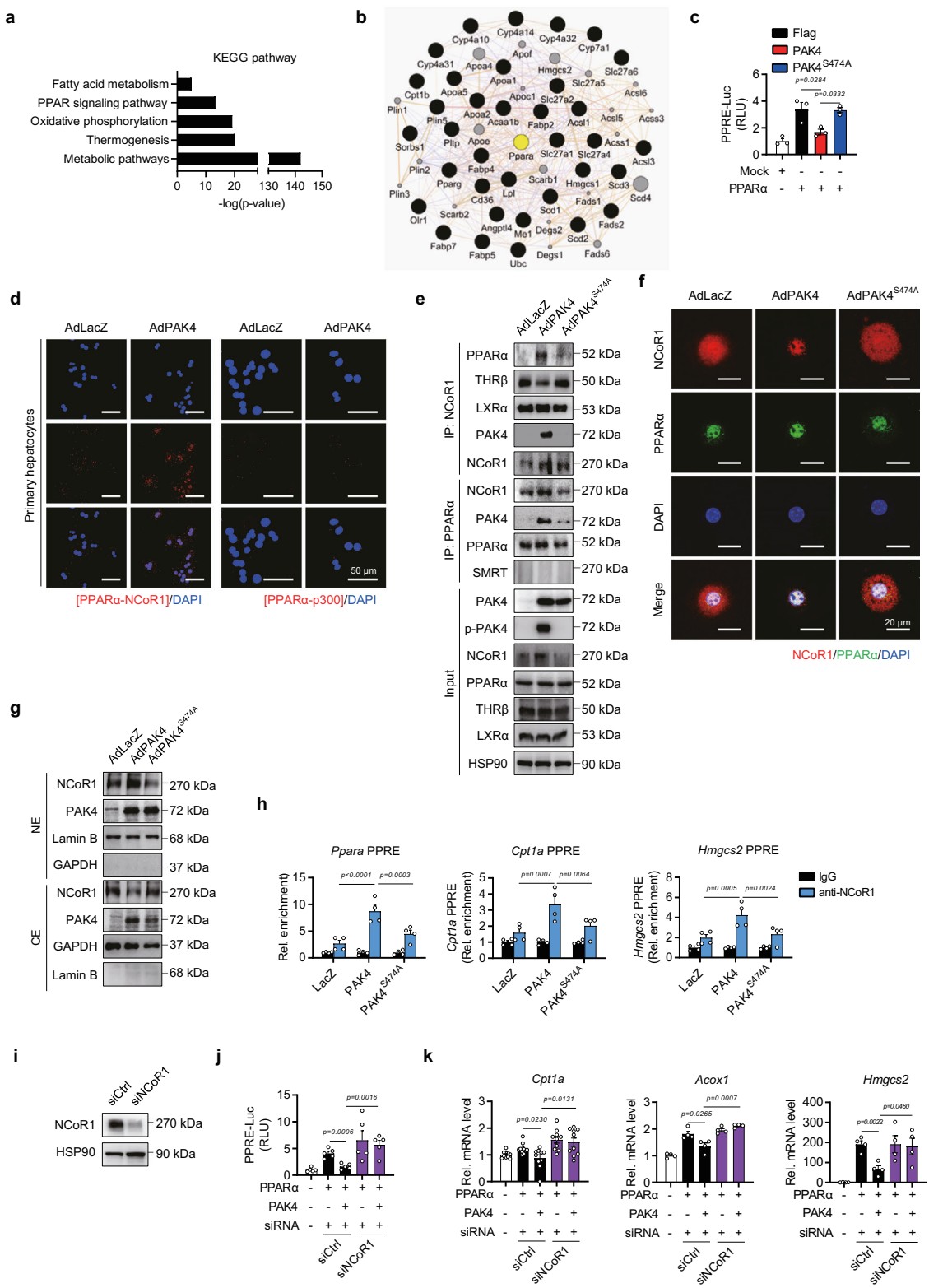

condition in WT mouse livers. Additionally, *Pak4* LKO mouse livers exhibited decreased phosphorylation, as well as reduced nuclear and total levels of NCoR1, even under fed conditions. As the subcellular localization of NCoR1 is linked with its phosphorylation status, and several kinases have been implicated, we examined these kinase signaling pathways. Importantly, we found that the dephosphorylation of NCoR1 in *Pak4* LKO is not associated with the reduced activation of either mTORC1/S6 kinase 2 (S6K2)[6,9] or Akt[7], which are known kinases for this corepressor (Fig. 5g). These results demonstrate that PAK4

directly phosphorylates NCoR1 on T1619/T2124 and enhances its gene-repressive function by preventing its nuclear export and the subsequent ubiquitination and proteasomal degradation.

We then overexpressed WT or double-mutant NCoR1 in mouse livers using adenoviruses. The PAK4-induced changes, including decreased fasting blood βOHB levels and increased liver TG contents, observed with WT NCoR1 overexpression, were attenuated in mice expressing the T1619A/T2124A mutant (Fig. 5h–j and Supplementary Fig. 6b, c). Furthermore, PAK4 overexpression led to a slight but

**Fig. 4 | PAK4 suppresses PPARα transcriptional activity by physically interacting with PPARα and NCoR1. a, b** KEGG pathway analysis of the DEGs from RNA-Seq (**a**) and GeneMANIA analysis showing an association of PPARα with the identified PPAR signaling pathway genes (**b**). **c** AML12 cells were co-transfected with PPARα expression plasmid, a firefly luciferase gene under the control of three tandem PPAR response elements (PPRE), and the plasmids encoding PAK4 or PAK4[S474A]. PPRE-luciferase activities were measured and expressed as the fold change relative to Mock (*n* = 3). **d–g** Primary hepatocytes were infected with the adenoviruses expressing control (AdLacZ), PAK4 (AdPAK4), or kinase-inactive mutant PAK4[S474A] (AdPAK4[S474A]), as indicated. PPARα binding to NCoR1 or p300 was analyzed with an in situ proximity ligation assay (PLA, **d**). Red fluorescent spots indicate that PPARα and NCoR1 are closely interacting (scale bars, 50 μm). Cell lysates were immunoprecipitated with the antibodies against NCoR1 or PPARα, followed by immunoblotting with anti-PPARα, anti-THRβ, anti-LXRα, anti-PAK4, anti-NCoR1, and anti-SMRT antibodies (**e**). Representative confocal microscopy images showing an increase of NCoR1 in the nucleus by PAK4 (**f**, scale bars, 20 μm). Nuclear (NE) and cytosolic extracts (CE) were analyzed for NCoR1 (**g**). **h** ChIP-qPCR assay assessing NCoR1 recruitment to the PPRE promoter regions of *Ppara*, *Cpt1a*, and *Hmgcs2* in mouse livers (*n* = 4). **i–k** AML12 were transfected with siRNA against NCoR1 (siNCoR1) or scrambled RNA (siCtrl) for 24 h, followed by transfection with the plasmids as indicated. Immunoblot images for NCoR1 (**i**), PPRE-luciferase activities (**j**, *n* = 5), and qPCR analysis (**k**) for *Cpt1a* (*n* = 10), *Acox1*, and *Hmgcs2* (*n* = 4). Cpt1a carnitine palmitoyltransferase 1a, Acox1 acyl-CoA oxidase 1, Hmgcs2 3-hydroxy-3-methylglutaryl-CoA synthase 2, RLU relative luminescence unit. Data are presented as the mean ± SEM. One-way ANOVA followed by Tukey's multiple comparisons test (**c**, **h**) and unpaired two-tailed *t* test (**j**, **k**) were conducted for statistical analyses. Source data are provided as a Source Data file.

significant increase in nuclear-NCoR1 in mouse livers injected with WT NCoR1 (Supplementary Fig. 6d), consistent with in vitro findings (Fig. 4f, g), but this effect was not observed with the mutant NCoR1 (T1619A/T2124A). These results clearly demonstrate that PAK4 regulates ketogenesis and liver TG accumulation through phosphorylation of NCoR1 at T1619/T2124.

### PAK4 deficiency restores an impairment of ketogenesis in HFD-fed mice

In line with our findings of increased PAK4 expression in mice with fatty liver (HFD-fed, *ob/ob*, and *db/db* mice, Fig. 1c), we also observed defective ketogenesis in these mice (Supplementary Fig. 7a). Building upon these findings, we proceeded to investigate the impact of PAK4 on ketogenesis in mice subjected to HFD feeding. After 5-week-HFD feeding, in comparison to WT mice, *Pak4* LKO mice in fasting condition displayed less TG accumulation in liver, higher βOHB levels in blood, and less nuclear- and p-NCoR1 in liver tissues (Supplementary Fig. 7b–e). In HFD-fed LKO mice, the level of Akt phosphorylation in the liver was higher compared to HFD-fed WT mice (Supplementary Fig. 7f). There were no differences in serum insulin levels and the Homeostatic Model Assessment of Insulin Resistance (HOMA-IR) index between the genotypes (Supplementary Fig. 7g, h).

In a similar context, liver tissues from NAFLD patients exhibited higher protein levels of PAK4 and NCoR1, along with decreased hepatic βOHB concentrations, in comparison to healthy subjects (Supplementary Fig. 7i, j).

### Small molecule inhibitor of PAK4 recapitulates *Pak4* LKO effects on ketogenesis

Next, we investigated the impact of a pharmacological inhibitor of PAK4, ND201651[19], on ketogenesis in mice. Male C57BL/6 mice aged 8 weeks were given a daily oral dose of ND201651 (50 mg/kg) for 14 days (Fig. 6a). ND201651 administration in KD-fed mice resulted in increased blood βOHB levels, decreased body weight, lowered blood glucose levels, and reduced TG content in the liver (Fig. 6b–d and Supplementary Fig. 8a). Notably, treatment with ND201651 did not induce toxicity, as indicated by decreased serum levels of aspartate aminotransferase (AST) and alanine aminotransferase (ALT), as well as unchanged tissue weights of major organs (EAT, gastrocnemius muscle, and kidney) (Supplementary Fig. 8b, c). Consistent with the results observed in *Pak4* LKO mice (Fig. 3k), treatment with ND201651 upregulated mRNA levels of *Ppara* and its target genes, *Cpt1a*, *Acox1*, and *Hmgcs2*, in KD-fed mice (Fig. 6e). In vitro experiments using hepatocyte cultures also confirmed that ND201651 increased PPRE-luciferase activity and PPARα target gene expression (Supplementary Fig. 8d, e). Furthermore, ND201651 completely abolished NCoR1 phosphorylation in AML12 cells (Supplementary Fig. 8f).

To further validate the therapeutic potential of the PAK4 inhibitor in treating ketogenesis-related diseases, we administered ND201651 (at doses of 25 and 50 mg/kg) to mice fed a HFD during the last 8 weeks of a 12-week feeding period (Fig. 6f). ND201651 administration resulted in increased blood levels of βOHB, reduced liver fat accumulation, and decreased NCoR1 phosphorylation (Fig. 6g–j).

Given the crucial role of FGF21 in ketogenesis through a mechanism partially independent of PPARα[3], along with the regulatory effect of PAK4 on FGF21 expression (Figs. 2f, i and 3g), we explored the potential involvement of FGF21 in the role of PAK4. Following ND201651 treatment, there was a notable increase in blood levels of βOHB and a decrease in glucose levels, accompanied by elevated protein levels of CPT1α and HMGCS2 in WT liver tissues (Supplementary Fig. 8g–i). However, these effects were nullified in *Ppara* KO mice. In addition, the increased production of ketone bodies in *Pak4* LKO hepatocytes remained unchanged even after silencing FGF21 (Supplementary Fig. 8j, k). These findings suggest that the inhibition of PAK4 specifically impacts the PPARα-dependent fatty acid β-oxidation pathway, leading to the production of βOHB, while implying that FGF21 does not exert a significant influence on these processes. Altogether, these studies provide compelling evidence for the therapeutic potential of the PAK4 inhibitor in treating fatty liver conditions associated with impaired ketogenesis.

### Enhanced ketone body levels in *Pak4* LKO mice inhibits tumor growth in mice

Since extracellular ketone bodies have been found to inhibit tumor growth through various mechanisms, including HDAC inhibition[20,33], we next determined whether hepatic PAK4 deficiency-enhanced ketogenesis suppresses tumor growth in in vivo tumor transplantation models; subcutaneous and orthotopic tumor models. In a subcutaneous tumor model, Hepa1-6 cells were subcutaneously implanted into WT and *Pak4* LKO mice, and tumor size was monitored while the animals were fed a NCD or KD (Supplementary Fig. 9a). We found that tumor mass was significantly lower in WT mice fed a KD than in those fed a NCD, without any changes in body weight (Supplementary Fig. 9b–e). Moreover, KD-fed *Pak4* LKO mice exhibited reduced tumor formation compared to WT mice on the same diet. Blood and hepatic levels of βOHB were higher in *Pak4* LKO than WT mice, and tumor weights were inversely correlated with βOHB levels (Supplementary Fig. 9f–i). In an orthotopic tumor model, Hepa1-6 cells were directly injected into the livers of WT and *Pak4* LKO mice (Supplementary Fig. 9j). Similar to the subcutaneous tumor model, mice fed with a KD exhibited reduced tumor formation with higher βOHB levels (Supplementary Fig. 9k–q). These effects were even more pronounced in *Pak4* LKO mice. These findings suggest that circulating or intrahepatic ketone bodies regulated by PAK4 play a crucial inhibitory role in tumor growth.

### PAK4 expression in tumor tissues of HCC patients inversely correlates with ketone body production

We analyzed liver tissues from HCC patients to investigate the association between PAK4 expression and the degree of ketogenesis, in

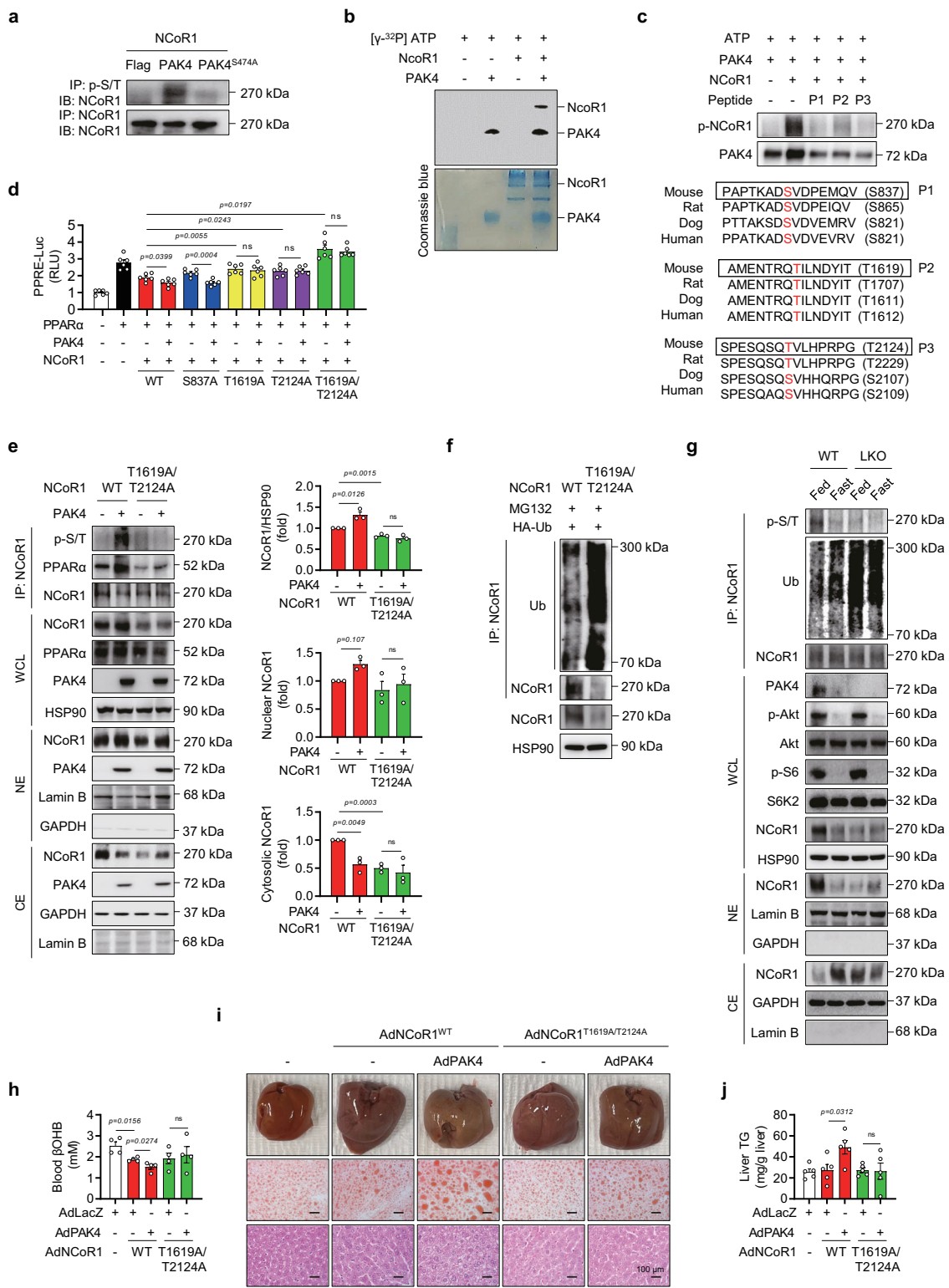

addition to NAFLD patients. Typical immunohistochemistry (IHC) staining for PAK4 and HMGCS2 in HCC tissues is presented in Supplementary Fig. 10a. The receiver operating characteristic curve analysis yielded a cut-off point of 7 for both PAK4 and HMGCS2. Cases where IHC staining scores were equal to or greater than 7 were considered positive for PAK4 and HMGCS2 immunostaining (Supplementary Fig. 10b). The factors significantly associated with PAK4 expression at these cut-off points were liver cirrhosis ($P = 0.034$) and tumor stage ($P = 0.023$) (Supplementary Table 1). No factors were

significantly associated with HMGCS2 expression. In the univariate analysis, consistent with previous reports that PAK4 is an indicator of aggressive tumor behavior and potentially affects patient outcomes[17,34,35], PAK4 and HMGCS2 expression was significantly associated with relapse-free (RFS) and overall survival (OS) (Supplementary Table 2). In the multivariate analysis, PAK4 and HMGCS2 expression independently predicted OS in HCC patients (Supplementary Table 2). The Kaplan–Meier survival curves indicated significantly poorer RFS ($P < 0.001$) and OS ($P < 0.001$) in HCC patients with high expression

**Fig. 5 | PAK4 phosphorylates NCoR1. a** AML12 cells were transfected with the plasmids encoding PAK4 or PAK4$^{S474A}$, along with NCoR1, as indicated. Phosphorylation of NCoR1 was analyzed by immunoblotting of NCoR1 after immunoprecipitation with a p-Ser/Thr antibody. **b** Recombinant NCoR1 was incubated with active PAK4 and [$^{32}$P] ATP for 30 min at 31 °C. Proteins in the mixture were resolved by SDS-PAGE and visualized by autoradiography. Loading of proteins was confirmed by Coomassie blue staining. **c** In vitro peptide-competing kinase assay using synthetic oligopeptides comprising putative PAK4 target sites (P1: S837, P2: T1619, or P3: T2124) was performed. The diagram shows the sequence alignment of the putative phosphorylation sites of NCoR1 in various mammalian species. **d** PPRE-luciferase activities were measured in HEK293T cells co-transfected with PAK4, PPARα, and either wild-type (WT) or mutated NCoR1 (S837A, T1619A, T2124A, or T1619A/T2124A) ($n = 6$). **e** Co-IP and immunoblot analysis in HEK293T cells after co-transfection of NCoR1 (WT or T1619A/T2124A) and PAK4. Whole-cell lysates (WCL), nuclear extracts (NE), and cytosolic extracts (CE) were analyzed for NCoR1 protein levels and quantified ($n = 3$). **f** Ubiquitination of NCoR1 in HEK293T cells after co-transfection of HA-Ub and NCoR1 (WT or T1619A/T2124A). **g** Phosphorylation and ubiquitination of NCoR1 were analyzed in liver tissues from fed- or 24 h-fasted *Pak4* LKO or WT mice. Protein levels of the indicated proteins were also analyzed in the WCL, NE, and CE. **h–j** Eight-week-old male C57BL/6 mice were injected with either adenoviruses expressing NCoR1 (AdNCoR1$^{WT}$ or AdNCoR1$^{T1619A/T2124A}$), PAK4 (AdPAK4), or LacZ (AdLacZ), and were subsequently fasted for 24 h. Blood levels of βOHB (**h**, $n = 4$), gross liver morphology, and microscopic analysis (Oil Red O and H&E stain) of liver tissues (**i**, scale bars, 100 μm), as well as TG levels in the liver (**j**, $n = 5$), were compared. RLU relative luminescence unit, ns no significance. Data are presented as the mean ± SEM. Unpaired two-tailed *t* test was conducted for statistical analyses (**d, e, h, j**). $^*P < 0.05$ and $^{**}P < 0.01$. Source data are provided as a Source Data file.

levels of PAK4 (Fig. 7a, b). In contrast, patients with high expression levels of HMGCS2 had significantly better OS ($P = 0.015$) and RFS ($P = 0.046$) rates than the low-expression group. Consistent with animal studies, immunoblotting analysis of liver tissues obtained from HCC patients indicated higher levels of PAK4, total-, phospho-, and nuclear-NCoR1, and lower levels of HMGCS2 in tumor compared to non-tumorous tissues (Fig. 7c and Supplementary Fig. 10c, d). Hepatic βOHB levels were inversely correlated with PAK4 expression in tumors (Fig. 7d), providing further evidence for the regulation of ketogenesis by PAK4 in the human liver.

## Discussion

Using physiologic (fasting) and nutritional (KD-feeding) ketosis models, we have uncovered distinct functions and mechanisms of PAK4 in the regulation of ketogenesis. We found that PAK4 deficiency promotes fatty acid β-oxidation and ketogenesis and subsequently attenuates hepatic fat accumulation, while PAK4 overexpression has the opposite effect. Treatment of mice with the PAK4 inhibitor ND201651 consistently increases ketogenesis and reduces hepatic fat content. In human livers obtained from NAFLD and HCC patients, PAK4 protein levels and NCoR1 phosphorylation show an inverse relationship with hepatic βOHB levels. Our findings extend the repertoire of regulators of NCoR1. Although several kinases have previously been reported, PAK4-dependent phosphorylation of NCoR1 on T1619/T2124 has not been described before. Our observation that phosphorylation at these sites of NCoR1 results in the repression of PPARα transcriptional activity through increased nuclear localization and interaction with PPARα highlights the importance of this newly discovered function of PAK4 (Fig. 7e).

We were surprised to discover that PAK4 protein levels were markedly repressed under fasting conditions. It is worth mentioning that PAK4 is often overexpressed in cancer cells and highly induced during oxidative stress[17,19]. These contrasting observations imply that PAK4 is sensitively regulated in response to environmental cues. We found that PAK4 undergoes ubiquitin-proteasome degradation, with three lysine residues (K31, K540, or K546) playing a critical role, and MDM2 being closely involved. This degradation occurs following phosphorylation at S258 by PKA activated by glucagon. Moreover, it has been demonstrated that Sirt1 also mediates the ubiquitin-proteasome degradation of PAK4. Specifically, the treatment of hepatocytes with ketogenic substrates (such as octanoate and palmitate) or βOHB itself enhanced the expression of Sirt1, resulting in the deacetylation and degradation of PAK4. These results suggest that PAK4 puts a brake on hepatic ketogenesis under fed conditions and that this brake is released upon fasting or starting ketogenesis, in the following sequence of events: fasting → glucagon/PKA activation and βOHB/Sirt1 induction → PAK4 degradation → NCoR1 de-repression → PPARα transactivation → βOHB production.

The gene-repressive function of NCoR1 is largely determined by its phosphorylation and nuclear localization. In the fed state, the protein is present in both the cytoplasm and nuclei in hepatocytes, and the nuclear form interacts with PPARα, which represses ketogenic gene expression[8]. Nuclear protein S6K2 has been reported to phosphorylate NCoR1 and lead to its accumulation in the nuclei[9]. mTORC1, an upstream molecule of S6K2, has been shown to be required for ketogenesis suppression through NCoR1[6], but some later studies have failed to observe a defect in ketogenesis in mice with hyperactivated mTORC1[36,37]. Akt phosphorylates NCoR1 and switches its binding target from lipogenic LXRα to PPARα for blunting ketogenesis during fed states[7]. However, there is contradicting evidence suggesting that Akt-mediated NCoR1 phosphorylation results in the redistribution of NCoR1 to the cytoplasm in neural stem cells[38]. During fasting, deactivation of mTORC1/S6K2 and Akt induces the release of NCoR1 from PPARα, which leads to the reactivation of PPARα[6,7,9]. It is still unclear how cytosolic NCoR1 is relocated to the nucleus in the fed state. In this study, we observed that PAK4 induced the upregulation of NCoR1 and its accumulation in the nucleus while decreasing its cytosolic level (Fig. 4e–g). In contrast, PAK4 deficiency resulted in the ubiquitin-proteasomal degradation and cytosolic localization of NCoR1, with no alterations in mTORC1/S6K2 or Akt signaling (Fig. 5g). We therefore suggest that PAK4 acts as a cue for the phosphorylation of cytosolic NCoR1, initiating its movement to the nucleus and facilitating additional phosphorylation by other NCoR1 kinases. Additionally, we propose that NCoR1 phosphorylation by PAK4 prevents its ubiquitin-proteasomal degradation, synergizing with nuclear localization to enhance its repressive function.

Additionally, PhosphoNET prediction, LC-MS/MS analysis, a peptide competition kinase assay, and phospho-inactive mutant study revealed that PAK4 is able to repress PPARα transactivation by phosphorylating NCoR1 on T1619 and T2124. Concordant with the fact that T2124 is located in the receptor-interacting domains of NCoR1[39], PAK4 phosphorylation of NCoR1 led to its association with PPARα. The significance of NCoR1 phosphorylation on T1619 and T2124 was further confirmed in mice. Overexpression of NCoR1 mutant (T1619A/T2124A) completely abolished the PAK4's effect of suppressing βOHB levels and increasing hepatic TG accumulation. Although it remains an open question how T1619 residue in NCoR1 affects its interaction with PPARα from a structural point of view, our NCoR1 mutant study highlights the importance of the phosphorylation of these residues for the functionality of NCoR1. Together, this study establishes a distinct role for PAK4 in the regulation of NCoR1/PPARα axis and ketogenesis. Considering the pleiotropic roles of NCoR1, future studies are warranted to validate the notion of PAK4 as an NCoR1 kinase in various disease models.

We also discovered that PAK4 inhibition can attenuate hepatic fat accumulation, by enhancing fatty acid β-oxidation and ketogenesis, which correlates with reduced phosphorylation of NCoR1. Of note, we present evidence that PAK4 inhibition-enhancement of fatty acid β-oxidation is the major driver of the ketogenesis and the consequent improvement of fatty liver. Firstly, the increase in βOHB

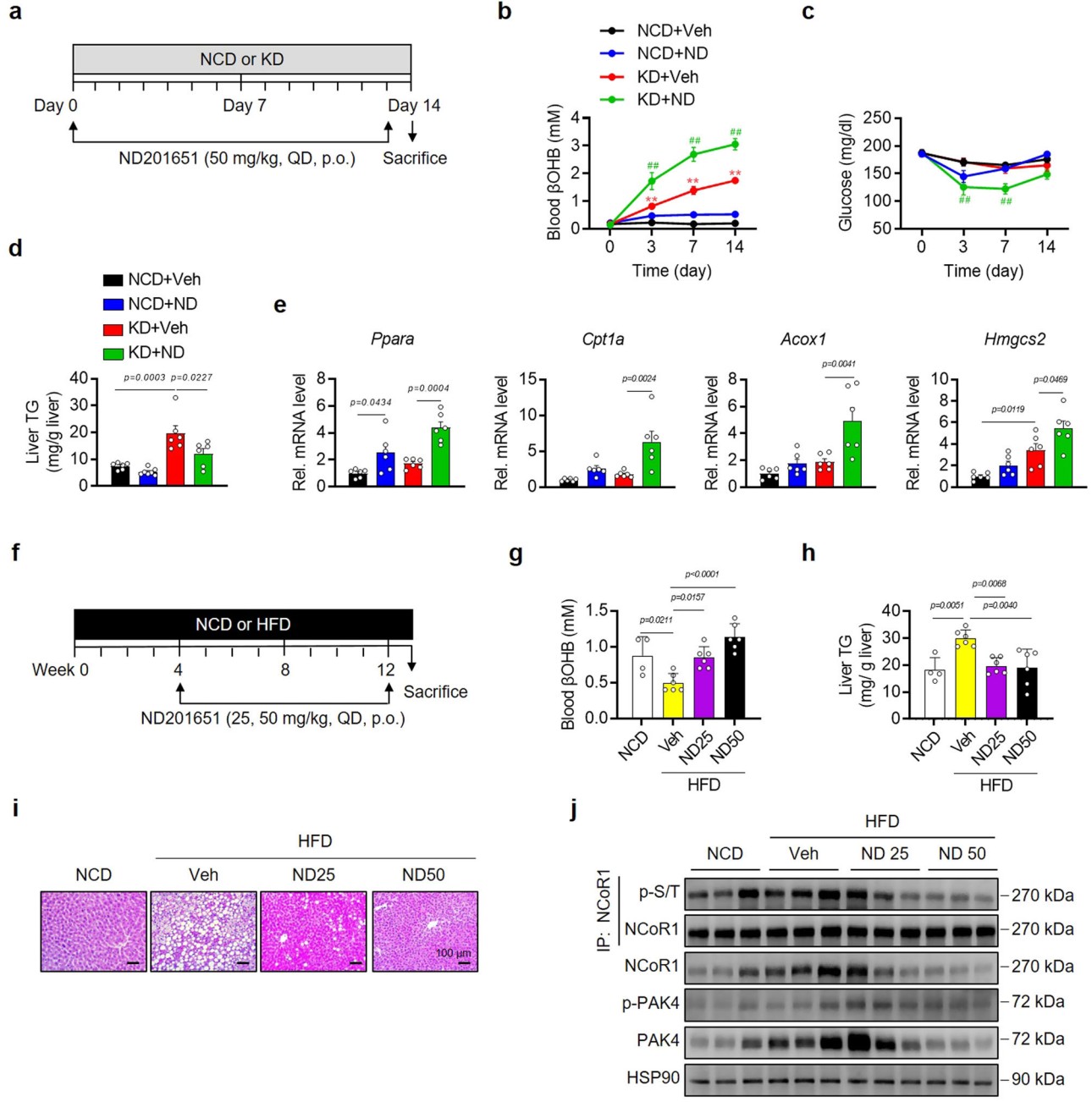

**Fig. 6 | PAK4 inhibitor ND201651 enhances ketogenesis in mice. a, f** Schematic diagrams illustrating the drug administration and the diet regimen in eight-week-old male C57BL/6 mice. Mice were orally given ND201651 (ND) once daily (QD) for 14 days (**a**) or 8 weeks (**f**) while fed a normal chow (NCD), ketogenic diet (KD), or high-fat diet (HFD). **b, c** Fed blood levels of βOHB (**b**, $n = 7$) and glucose (**c**, $n = 7$) were monitored at the indicated time points. $^{**}P < 0.01$ versus NCD+Veh; and $^{\#\#}P < 0.01$ versus KD+Veh. **d, e** Hepatic TG contents (**d**, $n = 6$) and qPCR analysis of β-

oxidation and ketogenesis genes (**e**, $n = 6$) were assessed on day 14. **g–j** Blood levels of βOHB (**g**), hepatic TG levels (**h**; **g, h**, $n = 4$ for NCD and $n = 6$ for HFD), hematoxylin & eosin (H&E) staining of liver sections (**i**, scale bars, 100 μm), and Western blotting analysis for the indicated proteins in liver tissues (**j**). Data are presented as the mean ± SEM. One-way ANOVA followed by Tukey's multiple comparisons test (**b–e, g, h**) were conducted for statistical analyses. Source data are provided as a Source Data file.

synthesis induced by the PAK4 inhibitor was nullified in *Ppara* KO mice (Supplementary Fig. 8g). This contrasts with FGF21, which promotes ketogenesis partly through a PPARα-independent mechanism[40]. Secondly, silencing FGF21 did not alter the elevated βOHB production observed in *Pak4* LKO hepatocytes (Supplementary Fig. 8j). This suggests that, although *Pak4* LKO mice exhibited increased FGF21 levels under fed conditions mimicking fasting (Fig. 3g), likely through PPARα activation, FGF21 did not play a critical role in mediating the ketogenesis induced by PAK4 inhibition. Interestingly, mice that overexpress PAK4 exhibited increased hepatic TG levels during fasting or on a KD, phenocopying mice lacking PPARα[3]. In contrast,

*Pak4* LKO mice or mice treated with a PAK4 inhibitor exhibited attenuated fat accumulation in the liver during fasting, on a KD, or HFD. Consistent with this, the levels of inflammatory genes and endoplasmic reticulum stress proteins were reduced in KD-fed *Pak4* LKO mice compared to WT mice. Our results collectively suggest that inhibiting PAK4 might be a therapeutic target for preventing hepatic steatosis.

The translational relevance of our study is further highlighted by our observations in patients with HCC. Firstly, protein levels of PAK4 and HMGCS2 were found to be independent and distinct predictors of survival in these patients. Secondly, hepatic levels of βOHB were

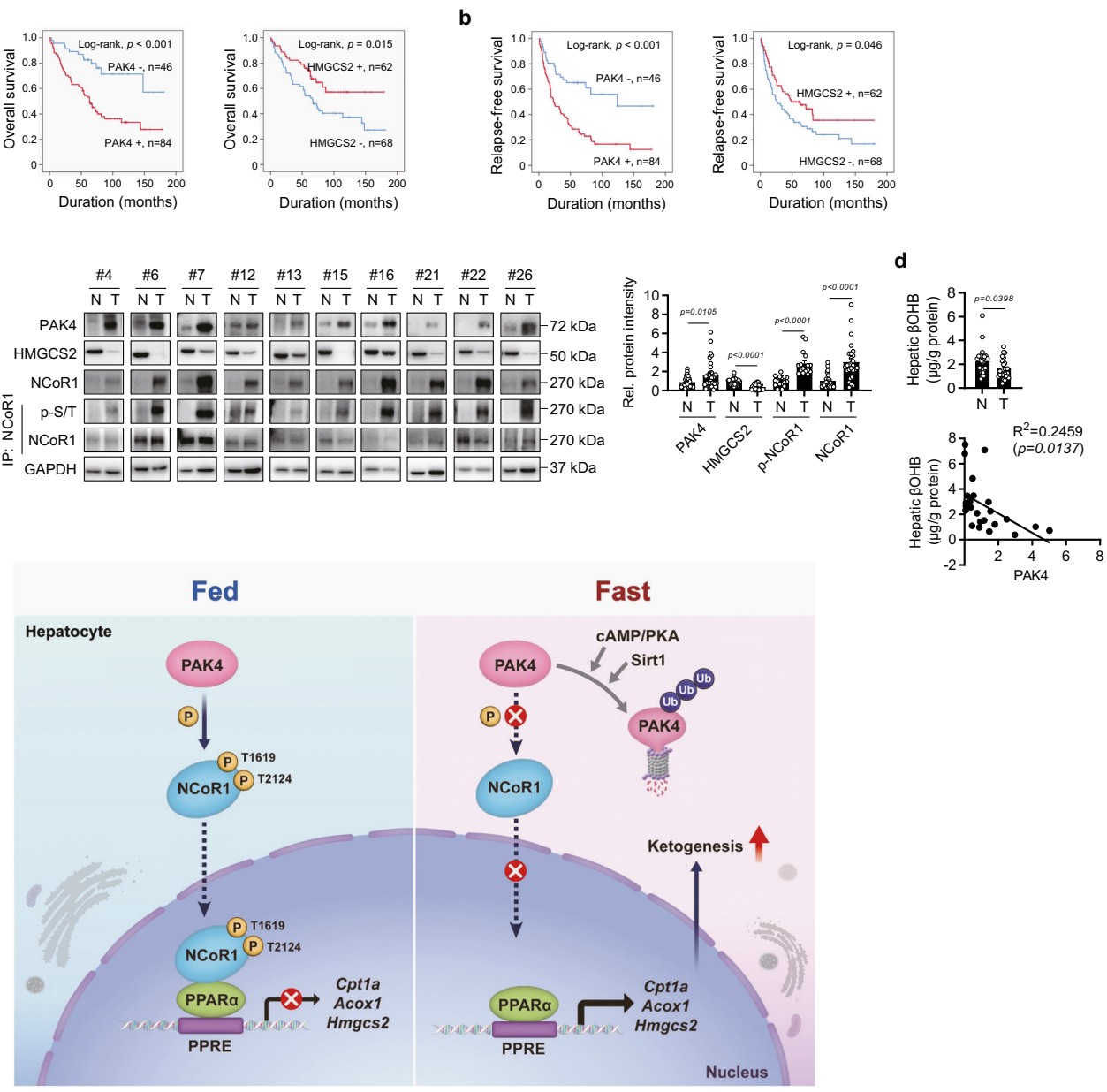

**Fig. 7 | Tumor overexpression of PAK4 is associated with reduced ketogenesis in hepatocellular carcinoma (HCC) patients. a, b** Kaplan–Meier survival curves are shown for overall survival (OS, **a**) and relapse-free survival (RFS, **b**) based on the expression of PAK4 and HMGCS2 in HCC patients. **c** Immunoblot analysis was performed for the indicated proteins in non-tumor (N) and tumor (T) liver tissues obtained from HCC patients ($n = 30$). NCoR1 phosphorylation was analyzed by immunoblotting with anti-p-Ser/Thr antibody following immunoprecipitation with anti-NCoR1 antibody. Protein density was quantified. **d** βOHB levels were measured in non-tumor and tumor liver tissues ($n = 23$). **e** The proposed summary is presented. Data are presented as the mean ± SEM. The survival of HCC patients was analyzed with univariate and multivariate Cox proportional hazards regression analyses and Kaplan–Meier survival analysis (**a, b**). Unpaired two-tailed $t$ test was conducted for statistical analyses (**c, d**). Pearson correlation coefficients were calculated between continuous variables (**d**). Source data are provided as a Source Data file.

negatively correlated with PAK4 protein levels. Thirdly, NCoR1 was highly phosphorylated in tumor tissues, in line with the enhanced PAK4 protein levels. Importantly, we found that higher and lower expression levels of PAK4 and HMGCS2 in tumor tissues in HCC patients were associated with poorer survival rates, consistent with previous reports showing that aberrant expression of PAK4 is associated with tumor progression or invasion[17] and that low levels of HMGCS2 and ketone bodies correlate with disease progression in HCC[41]. In line with this notion, HCC cells are known to exhibit reduced ketogenesis, redirecting from ketone production to ketolysis to furnish their own energy needs[15,42]. Considering that cancer cells undergo metabolic reprogramming in order to facilitate rapid cell proliferation[34], our findings that the

oncogenic protein PAK4 inhibits ketone production seem to be in line with its role in maintaining tumor growth.

There are a few limitations to this study. Firstly, although we identified three critical lysine residues (K31, K540, and K546) out of the four present in PAK4 that are involved in ubiquitin-proteasomal degradation, the detailed mechanism of PAK4 ubiquitination and degradation remains to be clarified. Secondly, only extrahepatic and intrahepatic tumor transplantation models were used in this study to investigate the role of PAK4 in regulating ketogenesis and its impact on tumor growth. Considering the limitations of traditional xenograft models for studying liver cancer, it may be necessary to employ spontaneous liver cancer models or humanized mouse models to

establish a more robust translational implication of ketone bodies in tumor growth. Lastly, although the results from our human study suggest a positive correlation between HMGCS2 and OS, we have yet to determine whether there is a causal relationship between local or systemic levels of ketone bodies and a good prognosis. Taken together, the present study shows that PAK4-mediated NCoR1 phosphorylation attenuates ketogenesis via PPARα repression under fed conditions or in fatty liver or HCC cells. However, in the fasted state or in normal hepatocytes, PAK4 protein is easily degraded, which leads to de-repression of PPARα-mediated ketogenesis. Therefore, PAK4 inhibition may be a new therapeutic strategy for pathogenic conditions associated with ketogenesis deregulation, such as NAFLD and HCC, owing to its ability to enhance the ketogenic capacities of the liver.

## Methods

### Institutional review board statement

Human study obtained institutional review board approval from Jeonbuk National University Hospital (IRB number CUH 2021-05-029, approved on 18 June 2021) and was conducted in accordance with the Declaration of Helsinki. The IRB approval included a waiver for obtaining written informed consent from patients for the use of tissue samples based on the retrospective and anonymous nature of the study. The biospecimens and data used in this study were provided by the Biobank of Jeonbuk National University Hospital, a member of the Korea Biobank Network, which is supported by the Ministry of Health, Welfare and Family Affairs. All samples obtained from the Korea Biobank Network were collected with informed consent under institutional review board-approved protocols. All animal experiments were performed in accordance with the Guide for the Care and Use of Laboratory Animals published by the US National Institutes of Health (NIH Publication No. 85-23, revised 2011). The study protocol of mice was approved by the Institutional Animal Care and Use Committee of Jeonbuk National University (permit number: JBNU-2019-00122).

### Immunohistochemical and immunofluorescence staining of tissue samples of HCCs

The data from 130 patients with HCCs, for whom histological slides and paraffin-embedded tissue blocks were available, were included in this study. All patients underwent radical resection at Jeonbuk National University Hospital between January 1998 and December 2009, and their clinical information was obtained by reviewing medical records. The cases included in this study were evaluated according to the latest World Health Organization Classification[43] and the staging manual of the American Joint Committee on Cancer[44]. For immunohistochemical staining of tissue samples, a tissue microarray containing one 3.0-mm core per HCC case was constructed. Tissue sections from the tissue microarrays were deparaffinized, and antigen retrieval was performed by boiling for 12 min in DAKO Target Retrieval Solution (pH 6.0; DAKO, Glostrup, Denmark) using a microwave oven. Thereafter, the tissue sections were incubated with primary antibodies for PAK4 (#SC390507, Santa Cruz Biotechnology, Dallas, TX, USA) and HMGCS2 (#D3U1A, Cell Signaling Technology, Danvers, MA, USA) and visualized using a DAKO Envision system (DAKO, Carpinteria, CA, USA). Intensity scores (0, no expression; 1, weak; 2, moderate; 3, strong) and area scores (0, no stained cells; 1, 1% stained; 2, 2–10%; 3, 11–33%; 4, 34–66%; 5, 67–100%) were added together[45]. The immunohistochemical staining score thus ranged from 0 to 8. The positivity of PAK4 and HMGCS2 immunostaining was then determined through a receiver operating characteristic curve analysis. The highest positive likelihood ratio points were used to predict the death of HCC patients[46]. In immunofluorescence staining, the slides were incubated with Alexa Fluor 594 anti-rabbit IgG (#A-11012, Invitrogen, Carlsbad, CA, USA) and counterstained with DAPI. The immunohistochemical staining slides for PAK4 and HMGCS2 were scored by two pathologists

(Drs. Kyu Yun Jang and Ho Sung Park) with consensus under a multi-viewing microscope.

RFS and OS rates were estimated for HCC patients through December 2013. The endpoint for the OS analysis was the death of the patient from HCC, and that for RFS was any relapse due to HCC or death. The survival of HCC patients was analyzed with univariate and multivariate Cox proportional hazards regression analyses and Kaplan–Meier survival curves. Associations between clinicopathologic variables were evaluated with Pearson's chi-square test. All statistical analyses were performed using SPSS software (IBM, version 20.0, CA, USA), and $P$ values < 0.05 were considered statistically significant.

### Animals and drug treatment

Eight- to ten-week-old mice were used in this study. C57BL/6 male or female mice were purchased from Damul Science (Daejeon, Korea). *Ob*/+, *ob*/*ob*, *db*/+, and *db*/*db* mice were obtained from Jackson Laboratory (Bar Harbor, ME, USA). *Pak4* LKO mice were generated by mating *Pak4*$^{flox/flox}$ mice (B6.129S2-*Pak4*$^{tm2.1Amin}$/J) with *Albumin-cre* (B6.Cg-*Speer6-ps1*$^{Tg(Alb-cre)21Mgn}$/J). *Pak4*$^{flox/flox}$ mice were used as littermate control mice. *Pparα* null mice were kindly provided by Dr. Goo Taeg Oh (Ewha Womans University, Seoul, Korea). Mice were maintained on a diet of standard laboratory chow (#31, Samtako, Seoul, Korea) and water *ad libitum* in a controlled barrier facility (12-h light/dark cycle, $23 \pm 1°C$, 60–70% humidity). Thereafter, they were fasted for 24 h or fed a KD containing 89.5% fat, 10.4% protein, and 0.1% carbohydrates (#D17011502, Research Diets, New Brunswick, NJ, USA) for 2 weeks. Adenoviruses carrying WT (AdPAK4) or a kinase-dead mutant of PAK4 (AdPAK4$^{S474A}$) were prepared as previously described[19]. To prepare PAK4 expressing adenovirus, the mouse PAK4 cDNA was inserted into pAdTrack-CMV vector (Addgene, Watertown, MA, USA). The D-TOPO-PAK4 plasmid was cloned into the pAdEasy-1 vector (Agilent Technologies, Santa Clara, CA, USA). PAK4 adenoviruses were injected into the mice's tail veins 3 days prior to fasting or onset of KD-feeding. Adenoviruses carrying either WT (AdNCoR1$^{WT}$) or a double-mutant of NCoR1 (AdNCoR1$^{T1619A/T2124A}$) were generated by Stratagene (La Jolla, CA, USA) and injected into the mice's tail veins alone or with AdPAK4, 3 days prior to fasting. *Pak4* LKO mice were fed with NCD or HFD (#D12492, Research Diets) for 5 weeks and fasted for 24 h before sacrifice. PAK4 inhibitor ND201651 was dissolved in 0.5% hydroxypropyl methylcellulose and administered to 8-week-old C57BL/6 male mice by oral gavage once a day for 14 days, along with NCD or KD-feeding. In another experiment, ND201651 was administrated to mice fed a HFD during the last 8 weeks of a 12-week feeding period at doses of 25 and 50 mg/kg every day. At the end of the experiments, the mice were terminated by $CO_2$ anesthesia.

### Bioluminescence imaging of the liver

Adenovirus expressing HMGCS2 (−231/+57) luciferase (AdHMGCS2-Luc) was generated using VectorBuilder (Guangzhou, China). AdHMGCS2-Luc ($1 \times 10^9$ pfu/kg of body weight) was injected into the tail veins of *Pak4* LKO or WT mice, followed by fasting for 24 h or KD-feeding for 3 days. Thereafter, mice were injected with 150 mg/kg of firefly D-luciferin (#LUCK-100, GoldBio, St Louis, MO, USA), anesthetized after 10 min, and imaged using the IVIS Luminar XR Imaging System (Caliper Life Sciences, Hopkinton, MA, USA).

### Cell culture, transient transfection, and promoter luciferase assay

Mouse primary hepatocytes were obtained as previously described[19]. Briefly, primary hepatocytes were isolated from *Pak4* LKO and WT mouse livers using collagenase type IV perfusion and 50% Percoll gradient centrifugation. The cells were plated in 6-well culture dishes at $1.5 \times 10^6$ cells/well in Medium 199 and used for Western blotting or ketone body production. The murine hepatocyte cell line AML12 (#CRL-2254), human embryonic kidney cell line HEK293T (#CRL-11268), and

murine hepatoma cell line Hepa 1–6 (#CRL-1830) were obtained from the American Type Culture Collection (Manassas, VA, USA). For PAK4 ubiquitination, AML12 cells were transfected with HA-Ub and PAK4 plasmids, with or without either siMDM2 (#156278, Ambion, Sydney, Australia) or siNEDD4 (#sc-41080, Santa Cruz Biotechnology) using Lipofectamine 3000 (Invitrogen). The cells were then treated with glucagon or forskolin for 12 h. For the knockdown experiment, siRNA (siSirt1, #93759; siHMGCS2, #15360; siFGF21, #56636; siNCoR1, #20185, all from Bioneer, Daejeon, Korea) was transfected into AML12 cells or primary hepatocytes using Lipofectamine RNAiMAX (Invitrogen), according to the manufacturer's instructions. The PPARα and FGF21 reporter gene assay was prepared using 0.5 μg of a plasmid containing a promoter with a PPRE. Exogenous proteins were expressed by transfecting AML12 cells with 1 μg of PPARα, PAK4, and PAK4$^{S474A}$. Cells were treated with the ND201651 24 h after transfection, and luciferase activity was measured using the Dual-Luciferase reporter assay system (#E1910, Promega, Madison, WI, USA).

## Subcellular fractionation, co-immunoprecipitation, and Western blotting

Tissues and cells were homogenized in Tissue Protein Extraction Reagent or Mammalian Protein Extraction Reagent (#78510 or 78505, Thermo Fisher Scientific, Waltham, MA, USA). Nuclear and cytoplasmic extracts were isolated using the NE-PER Nuclear and Cytoplasmic Extraction Kit (#78835, Thermo Fisher Scientific). Homogenates (20 μg for Western blot and 200 μg for co-IP) were separated by SDS-PAGE and transferred to nitrocellulose membranes. After blocking with 5% skim milk, blots were probed with primary antibodies (Supplementary Table 3) against PAK4, p-PAK4 (S474), HSP90, ubiquitin, HMGCS2, phospho-PKA Substrate, phospho-(Ser/Thr), Sirt1, Sirt6, phospho-Akt (Ser473), Akt, acetylated-Lysine, CHOP, phospho-S6 Ribosomal Protein (Ser240/244), S6 Ribosomal Protein, NCoR1, NEDD4, ATF-6, phospho-PERK (Thr980) (Cell Signaling Technology), PAK4, FGF21, and PPARα (Santa Cruz Biotechnology), Sirt4, lamin B1, and GAPDH (Bioworld Technology, St Louis Park, MN, USA), Sirt5, Sirt7 and FoxO1 (Acetyl-Lys294) (LifeSpan Biosciences, Seattle, WA, USA), and Sirt2, Sirt3, GRP78 BiP, NCOR2/SMRT, CPT1α or CPT1A,T-OXPHOS, THRβ, p-IRE1α and MDM2 (Abcam, Cambridge, UK), LXRα (Proteintech, Rosemont, IL, USA). Immunoreactive bands were detected with an LAS-4000 imager (GE Healthcare Life Science, Pittsburgh, PA, USA).

## RNA isolation, qPCR and genotyping

Total RNA was extracted from frozen liver tissues or primary hepatocytes using the RNA Iso Kit (#9109, TaKaRa, Tokyo, Japan). First-strand cDNA was generated using the random hexamer primer provided in the first-strand cDNA synthesis kit (#K1672, Thermo Fisher Scientific). Specific primers were designed using PrimerBank (https://pga.mgh.harvard.edu/primerbank, Supplementary Table 4). qPCR reactions were performed in a final volume of 10 μl containing 10 ng reverse-transcribed total RNA, 200 nM forward and reverse primers, and PCR master mixture. qPCR was performed on 384-well plates using an ABI Prism™ 7900HT Sequence Detection System (Applied Biosystems). For genotyping, reverse transcription and PCR were performed with a One-Step RT-PCR Kit (Invitrogen). PCR fragments were separated by electrophoresis on 2% agarose gels, followed by staining with ethidium bromide.

## Chromatin immunoprecipitation (ChIP)

A ChIP assay of liver lysates was conducted using the ChIP Enzymatic Chromatin IP Kit (#9003, Cell Signaling Technology). Liver tissues from mice fed a KD after adenovirus injection were lysed and cross-linked by incubating cells in 1% formaldehyde for 15 min at room temperature. Cross-linking was stopped by 5 min of incubation with 125 mM glycine. Chromatin was immunoprecipitated overnight at 4 °C with antibodies against NCoR1 or nonspecific IgG (Cell Signaling

Technology). Data were normalized to input quantity. All primer sequences are listed in Supplementary Table 4.

## Biochemical analysis

Blood glucose levels were measured at indicated time points using an Accu-Chek (Roche, Basel, Switzerland). Serum levels of glucagon (#48-GLUHU-E01) and insulin (#80-INSMS-E01) were measured by specific assay kits (both from ALPCO, Salem, NH, USA). Serum levels of free fatty acids were determined using a colorimetric assay kit (#MAK044, Sigma-Aldrich). Serum level of FGF21 was detected by a FGF21 ELSIA kit (#212160, Abcam). Tail-tip blood and medium βOHB levels were measured using the CareSens Dual monitoring system (i-SENS, Seoul, Korea). Hepatic βOHB levels were measured with a βOHB colorimetric assay kit (#700190, Cayman Chemical, Ann Arbor, MI, USA). Liver and serum TG levels were determined using a TG assay kit (#AM157S-K, Asan Pharmaceutical, Seoul, Korea). To quantify liver TG, liver tissues (100 mg) were homogenized and extracted in a 1-ml mixture of 5% Triton X-100 (Sigma-Aldrich) in PBS. Serum levels of ALT and AST were analyzed using specific kits (#K753-100 and K752-100, Biovision, Milpitas, CA, USA).

## Histology

Liver tissues were fixed in 10% formaldehyde solution at 4 °C for paraffin sections or placed in 30% sucrose solution and embedded with liquid nitrogen-cooled isopentane for frozen sections. Paraffin sections of liver tissue were stained with hematoxylin and eosin (H&E) for light microscopy. For Oil Red O staining, frozen sections were incubated with the dye for 15 min and washed with 1× PBS. For immunofluorescence analysis, primary hepatocytes were stained overnight at 4 °C with antibodies against NCoR1 (Cell Signaling Technology) or PPARα (Santa Cruz Biotechnology). After washing with PBS, the sections were incubated with secondary antibodies (Alexa Fluor 488-conjugated goat anti-mouse IgG1 and Alexa Fluor 594-conjugated goat anti-rabbit IgM; Thermo Fisher Scientific) for 1 h at 37 °C and then counterstained with DAPI. NCoR1 and PPARα interactions were detected using a Duolink Proximity Ligation Assay kit (#DUO96010, Sigma-Aldrich).

## RNA sequencing (RNA-Seq) analysis

RNA was extracted from liver tissues of WT and hepatocyte-specific *Pak4* LKO mice after 24 h of fasting. The total RNA concentration was measured using the Quant-IT RiboGreen assay (#R11490, Invitrogen). RNA integrity was evaluated using the TapeStation RNA ScreenTape system (#5067-5576, Agilent Technologies), and only samples with a RIN greater than 7.0 were used for RNA library preparation. The library was constructed independently for each sample using 0.5 μg of total RNA and the Illumina TruSeq Stranded Total RNA Library Prep Gold Kit (#20020599, Illumina). The first step in the library construction workflow involved removal of rRNA, followed by fragmentation of the remaining mRNA into small pieces using divalent cations at elevated temperatures. The cleaved RNA fragments were then converted into first-strand cDNA using SuperScript II reverse transcriptase (#18064014, Invitrogen) and random primers, followed by second-strand cDNA synthesis using DNA Polymerase I, RNase H, and dUTP. The resulting cDNA fragments underwent an end repair process, the addition of a single "A" base, and ligation of adapters. The final cDNA library was purified and enriched with PCR, and the libraries were quantified and qualified using KAPA Library Quantification kits for Illumina Sequencing platforms and the TapeStation D1000 ScreenTape (#5067-5582, Agilent Technologies). The indexed libraries were then submitted for paired-end (2 × 100 bp) sequencing on an Illumina NovaSeq (Illumina), performed by Macrogen (Seoul, Kora).

Relative gene abundances were measured in read counts using StringTie. A statistical analysis was performed to identify differentially expressed genes (DEGs) using the estimates of abundances for each

gene in the samples. Genes with more than one zeroed read count value were excluded. To facilitate log2 transformation, 1 was added to each read count value of filtered genes. Filtered data were $log_2$-transformed and subjected to TMM normalization. Statistical significance was determined using exactTest with edgeR, with fold changes, and the null hypothesis that no difference exists among the groups. The false discovery rate (FDR) was controlled by adjusting $P$ values using the Benjamini–Hochberg algorithm. For the DEG set, a hierarchical clustering analysis was performed using complete linkage and Euclidean distances as measures of similarity. Gene enrichment, functional annotation, and pathway analyses for significant gene lists were performed using gProfiler (https://biit.cs.ut.ee/gprofiler/gost) and the Kyoto Encyclopedia of Genes and Genomes (KEGG) pathway (http://www.genome.jp/kegg/pathway.html).

### Gene set enrichment analysis (GSEA)
RNA-Seq data were analyzed using GSEA 4.1.0 software and hallmark gene sets listed in the Molecular Signatures Database (http://software.broadinstitute.org/gsea/msigdb) v7.4. FDR was used for the statistical significance assessment of NES. Gene sets with FDR < 0.25 were considered statistically significant. Gene interaction networks were assessed with GeneMANIA (https://genemania.org/), and visualized using Cytoscape 3.8.2 software (http://www.cytoscape.org/).

### In vitro kinase assay
Recombinant human NCoR1 protein (0.5 μg, #ab82239, Abcam) was incubated with recombinant human PAK4 protein (0.2 μg, #ab96405, Abcam) in assay buffer (50 mM Tris-HCl at pH 7.6, 10 mM MgCl₂, 2 mM DTT, and 0.1 mM EDTA) containing 5 μCi of [γ-$^{32}$P] ATP and 50 μM cold ATP at 31 °C for 30 min. Reaction mixtures were then subjected to SDS-PAGE, and $^{32}$P-labelled proteins were detected by autoradiography. For Coomassie blue staining, the gel was stained in protein stain buffer (#ab119211, Abcam) for 1 h. For the oligopeptide competition assay, three different synthetic 15-residue oligopeptides mimicking the putative PAK4 target sites (P1, P2, and P3 comprising S837, T1619, or T2124, respectively) were used as competitors and incubated with human NCoR1 (0.5 μg) and PAK4 (0.2 μg) in kinase assay buffer containing cold ATP. End products were resolved on SDS-PAGE and detected by immunoblotting with p-Ser/Thr antibody.

### Mutagenesis of NCoR1
Mouse NCoR1 plasmid vector was kindly provided by Dr. Young Suk Jo (Yonsei University College of Medicine, Korea). The mutants of NCoR1 were generated by site-directed mutagenesis (Stratagene, La Jolla, CA, USA). NCoR1-S837A, NCoR1-T1619A, and NCoR1-T2124A mutants were generated by introducing a point mutation of the serine or threonine residue in NCoR1 to alanine as follows: NCoR1-S837A (TCC → GCC), NCoR1-T1619A (ACA → GCA), and NCoR1-T2124A (ACT → GCT).

### LC-MS/MS and peak alignment
The phosphopeptides were analyzed with a Q-Exactive HF-X mass spectrometer (Thermo Fisher Scientific) connected to UHPLC system (Ultimate 3000, Thermo Fisher Scientific) to acquire MS/MS spectra. The peptide samples were loaded into the trap column and separated on an analytical column (PepMap RSLC, 3 μm, 100 Å, 75 μm × 50 cm). Two μl of each sample was injected onto a binary mobile phase consisting of water in 0.1% formic acid as mobile phase A and acetonitrile in 0.1% FA as mobile phase B. A linear gradient consisting of 2–35% mobile phase B at a flow rate of 300 nl/min was employed for 70 min.

For phosphorylation of NCoR1 on T1619 and T2124 identification, peptides were analyzed using Parallel Reaction Monitoring (PRM). PRM were conducted under full scan and followed integrated method: nanospray source voltage 1.8 kV, capillary temperature 275 °C, max injection time 100 ms, and target value $1 \times 10^6$. Mass spectra was acquired between 375 and 1800 with a mass resolution of 60,000. Lock mass ion from polysiloxane ($m/z$ 445.12003) was applied. Target $m/z$ of phosphopeptide ion precursors were isolated after a full mass scan (as listed up in Supplementary Table 5), target value was $2 \times 10^5$, maximum free injection time was 64 ms at 30,000 resolution, loop count was 10, and isolation width was 1.6 $m/z$. MS was run on positive mode.

PRM spectra for phosphopeptide identification and its site was obtained in Mascot (version 2.8.0.1) database analysis server. The resulting data was used to query a database containing 55,275 entries of *Mus musculus* downloaded from UniprotKB (released on Feb, 2023). Search parameters were described as follows: trypsin/P digestion with up to two missed cleavages, acetylation (N-term), oxidation (Met), and phosphorylation (Ser/Thr/Tyr) as variable modifications. The search results were filtered with FDR of less than 1% with decoy mode, Peptide tolerance was set to 0.5 Da and MS/MS tolerance was 20 ppm for monoisotopic mass.

### Tumor transplantation models
For the subcutaneous tumor transplantation model, $1 \times 10^6$ Hepa 1–6 cells were resuspended in 100 μl Matrigel matrix (1:1 with PBS, # CLS354234, Thermo Fisher Scientific) and injected subcutaneously onto both flanks of 8-week-old male *Pak4* LKO or WT mice. Once a tumor had grown to 500 mm³, the tumor-bearing mouse was fed either an NCD or KD for 9 days. Tumor size was measured with electronic calipers and volumes were determined using the following formula: length (mm) × width (mm)² × 0.5. Mice were sacrificed when their tumors had grown to ~2000 mm³. For the orthotopic tumor transplantation model, $1 \times 10^6$ Hepa 1–6 cells were directly injected into the livers of WT and *Pak4* LKO mice. One week later, mice were fed with either an NCD or KD for an additional 2 weeks.

### Statistics and reproducibility
Data are expressed as the mean ± standard error of the mean (SEM). All $n$ values indicated by dots in the figures refer to biological replicates. All results are from at least three independent samples or experiments. The significance of differences between two groups was determined using two-tailed Student's unpaired $t$ tests. For comparisons among more than two groups, one-way ANOVA was used followed by Dunnett's multiple comparisons test or Tukey's multiple comparisons test. A $P < 0.05$ was considered significant. Statistical analyses were performed using GraphPad Prism 9.4 software (GraphPad Software, La Jolla, CA, USA).

### Reporting summary
Further information on research design is available in the Nature Portfolio Reporting Summary linked to this article.

### Data availability
Raw and processed RNA-seq datasets generated in this study have been deposited in the NCBI GEO database under accession code GSE214442. Source data are provided with this paper.

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

## Acknowledgements

We thank Dr. Sun-Shin Cha at TODD Pharmaceuticals for providing recombinant FGF21. This work was supported by grants from the

Medical Research Center Program (2017R1A5A2015061), Bio & Medical Technology Development Program (2022M3E5F2017607), the Korea Drug Development Fund (HN21C0447), and Basic Science Research Program (2023R1A2C3002389), all of which were funded by the Korean government.

## Author contributions

B.H.P. and E.J.B. conceived the idea, designed the experiments, and wrote the manuscript. M.Y.S., H.C.Y., C.Y.H., I.H.B., H.S.P., and K.Y.J. conducted the experiments and analyzed the data. S.L. performed the LC-MS/MS analysis. J.B.S. and N.D.K. synthesized ND201651. B.H.P. and E.J.B. have the primary responsibility for the content of the final version of the manuscript. All authors read the manuscript and approved its submission.

## Competing interests

The authors declare no competing interests.
