## [Peer Review File · Nature Communications]

p21-activated kinase 4 suppresses fatty acid β -oxidation and ketogenesis by phosphorylating NCoR1REVIEWER COMMENTS

Reviewer #1 (Remarks to the Author):

In this manuscript, Shi and colleagues investigated the role of PAK4-NCoR1/PPAR α signaling pathway in regulating ketogenesis during liver diseases. The authors found that PAK4 was increased in liver tissues of high fat diet-fed mice, NAFLD patients, and hepatocellular carcinoma patients. PAK4 phosphorylated NCoR1 on T1619/T2124 and induced its translocation towards nucleus to suppress PPAR α and ketogenesis. PAK4 protein levels were significantly suppressed by fasting through either cAMP/PKA- or Sirt1-mediated ubiquitination and proteasome degradation. They provided evidence for a PAK4-NCoR1/PPAR α signaling pathway regulating ketogenesis. However, this research lacks enough innovation, and the conclusion of some aspects lacks sufficient evidence.

Main Points:

1. In Figure 1, the authors found levels of PAK4 protein in liver tissues/cells were regulated by fatty acid metabolism via ubiquitin-mediated degradation of PAK4, but not via transcription. Here, the research is too superficial, the key point is to identify the exact ubiquitinated sites in PAK4. Additionally, the authors found p-S/T was associated PAK4 protein levels, what is the sites of p-S/T? is it PAK4 S474 or the other sites? Is proteasome-dependent degradation of PAK4 resulted from the Ser/Thr phosphorylation of PAK4?
2. PAK4 phosphorylates NCoR1 on T1619/T2124 according to the predictive tools, I think the authors should provide more rigorous approach such as IP-MS/MS to identify the phosphorylated sites in NCoR1.
3. β OHB acts as an endogenous inhibitor of histone deacetylases (HDACs) (PMID: 24140022). OCA results in increase of β OHB, which could inhibit activity of HDACs. Why the authors employed panobinostat (HDACs inhibitor) to rescue PAK4 decrease upon OCA treatment?
4. The author claimed that sirt1 is a regulator of PAK4, however, in fig.S1J sirt3 and sirt7 were unregulated and sirt4 is downregulated upon OCA treatment. Are there sirt3 and sirt7 play role in PAK4 stabilization? In addition, cAMP/PKA/SIRT1 axis has been well elucidated (PMID: 33045622, PMID: 33369003). Why the author claimed cAMP/PKA and SIRT1 could play independently in regulating stabilization of PAK4.

5. In Fig.4f, why not show PAK4 S474A group using IF. Additionally, the authors should test LaminB and GAPDH in all samples to confirm the quality of plasma and nucleus separation experiment.
6. In Fig. 5b, NCoR1 purified protein was not shown in CBB figure. In Fig.5F, in double mutant group, the total NCoR1 were decrease, which cannot conduct the decrease of NCoR1 nuclear translocation was induced by mutation of T1619A/T2124A. Besides, NCoR1 decreased significantly in fast and LKO group. Were NCoR1 were degraded through ubiquitin proteasome system under these conditions?
7. The data of PAK4 roles in liver cancer are not solid. In Supplementary Fig. 7C, the authors should provide the picture of tumors. In addition, the xenograft tumor model used here is not enough. The authors should use spontaneous model of liver cancer with KD or NCD diet to explore the function of PAK4. Is NCoR1 T1619/T2124 involved in this process?

Minor points

1. The data is not rigorous and lacks control/input groups in some figures. For example, the input group is missed in Fig.S1K.
2. GAPDH is an enzyme involved in glycolysis. Glucagon and insulin will alter the expression and activity of this enzyme. Therefore, it is not appropriate to use GAPDH as an internal parameter.
3. Lots of gene enriched in lipid metabolism were influenced in PAK4 LKO group, which may involve in regulating NAFLD pathology. Why the author put attention on ketogenesis and NCoR1?
4. The authors should analyze the nuclear localization of NCoR1 in high or low expression of PAK4 in human tissues by IF directly.
5. In GSEA, $FDR < 0.25$ was considered a trusted enrichment. Fig.3a, the FDR was 0.489 far from 0.25 in GSEA about Regulation of Ketone Biosynthetic Process. So not sure if it is relevant to Ketogenesis. It is best to display heat map of genes that enriched in this pathway and detect their expression.

Reviewer #2 (Remarks to the Author):

In their manuscript titled “p21-activated kinase 4 suppresses ketogenesis by phosphorylating NCoR1”, She and co-authors build a compelling story demonstrating that PAK4 kinase, previously described as an oncoprotein, phosphorylates nuclear corepressor NCoR1 leading to an increase of its nuclear localization that leads to inhibition of ketogenesis through transcriptional repression of PPAR alpha.

The authors demonstrate that levels of hepatic PAK4 are increased in mouse models of obesity and fatty liver, such as HFD-induced and genetically induced (Ob/ob and db/db) obesity. PAK4 protein is downregulated in fasting through two independent mechanisms (PKA and Sirt1) thus allowing appropriate physiological activation of ketogenesis during fasting.

Overexpression of PAK4 (but not the kinase-inactive mutant) leads to decreased expression of enzymes in fatty acid oxidation and ketogenic pathways and fat accumulation in the liver, particularly in conditions where ketogenesis is normally activated, such as fasting and low carb (ketogenic) diet feeding.

Liver-specific KO of NAK4 leads to enhanced expression of ketogenic enzymes in the liver, increased serum B-OHB and decreased hepatic and circulating TGs during fasting and on ketogenic diet. RNAseq analysis identified PPAR alpha as the key factor connecting the pathways affected by NAK4 KO. PAK4, but not the kinase-inactive mutant suppressed PPAR alpha in luciferase assays, but did not affect phosphorylation of PPAR alpha itself. Proximity ligation assays and Co-IPs identified nuclear receptor corepressor NCoR1 as the phosphorylation target of PAK4. This event increases nuclear localization of NCoR1 and facilitates its interactions with PPAR alpha leading to repression of its transcriptional activity. Increased binding of NCoR1 to PPAR alpha response elements by overexpression of PAK4 was also confirmed through CHIP.

Effects of PAK4 overexpression on ketogenesis were blunted by knock-down or overexpression of NCoR with mutated PAK4 phosphorylation sites.

The authors also noted decreased TG accumulation and increased circulating BOHB in the livers of mice with hepatic KO of PAK4 fed with HFD.

A small molecule inhibitor of PAK4 recapitulated effects of hepatic PAK4 KO on ketogenesis in mice fed ketogenic and high fat obesogenic diets.

Importantly, authors also demonstrate that increased ketogenesis through KO of hepatic

PAK4 decreases extrahepatic tumor growth in mice fed ketogenic diets.

WB analysis of biopsies material from human HCC demonstrated higher expression of PAK4 and NCoR1 and lower levels of HMGCS2 in tumor compared to non-tumor tissues. In line with these and previous finding in human HCC population, overall and relapse-free survival were significantly poorer in HCC patients with high expression levels of PAK4.

This is a well-written study with exciting new findings and potentially important ramifications for human health. The study is well designed, executed, and presented. The conclusions are well-supported by the data. The methodology is sound, and methods are described in sufficient detail. Limitations of the study are also acknowledged by the authors.

Below are some more general comments and suggestions.

1. The basic physiologic model proposed by the authors describes an elegant mechanism for additional control of beta-oxidation and ketogenesis, where NAK4 is abundant and active in fed conditions. It phosphorylates NCoR1 promoting its interactions with PPAR alpha to suppress ketogenesis in fed animals, where it is physiologically irrelevant. During fasting NAK4 levels drop allowing for de-phosphorylation of NCoR and its dissociation from PPAR alpha, so ketogenic gene expression program can be activated. However, the strongest phenotypic effects of hepatic NAK4 KO on liver and serum TGs, and circulating B-OHB are observed in conditions where ketogenesis is already active: in fasting and on ketogenic diet. mRNA and protein expression of relevant enzymes are also shown under these conditions. Can the authors elaborate on this apparent discrepancy between the basic model and the conditions where experimental observations were made?
2. Along the lines of the first point, it would be interesting to hear if the authors could comment on whether NAK4-mediated phosphorylation of NCoR1 also affects interactions of this corepressor with other nuclear receptors, whose functions are reciprocal to PPAR alpha (LXR, THR)?
3. Finally, the authors should be cautious when using term "nuclear translocation" describing effects of NAK4 on NCoR1, as there is no data in this manuscript that addresses the exact mechanism of the changes of intracellular localization of NCoR1.

Reviewer #3 (Remarks to the Author):

In this manuscript, the authors detail a mechanism through which p21 activated kinase 4 (PAK4) regulates ketogenesis through the interaction between nuclear receptor corepressor 1 (NCoR1) and PPAR α . The manuscript details elegant studies to identify the above interactions and the phosphorylation sites of NCoR1 leading to its impact on ketogenesis. While the detailed molecular mechanisms are elaborate and robust, the authors should address the following concerns.

1. All the metabolic impacts detailed in the manuscript clearly point to lipid oxidation (β -oxidation) rather than ketogenesis as the major driver of the observed effects in the liver. While the majority of ketogenesis results from break down of fatty acids, the mechanisms proposed in the manuscript seem to modulate the pathway of β -oxidation rather than ketogenesis. The authors should clarify the rationale for focusing on ketogenesis rather than lipid oxidation. If the proposed mechanisms have a specific impact on ketogenesis, wouldn't it be also logical to explore the PPAR α mediated regulation of FGF-21 in these experiments? The authors should clearly delineate the impacts of the PAK4-NCoR1/PPAR α axis on β -oxidation vs. ketogenesis.
2. While the broad idea of correcting defects in ketogenesis/lipid oxidation through the modulation of the PAK4-NCoR1/PPAR α axis is attractive, the proposed mechanisms will induce mitochondrial activity. The impacts of chronic induction of ketogenesis/lipid oxidation on mitochondrial activity/OXPHOS, hepatocellular stress/ inflammation is not explored, especially in the context of evaluating the therapeutic potential of these mechanisms.
3. The broad theme of the manuscript investigates the impact of the proposed mechanisms mediated by PAK4 on lipid accumulation in the liver. Towards determining clearly, the beneficial impact of the proposed mechanisms on lipid metabolic network in the liver, estimates of changes in insulin sensitivity/ signaling (e.g., AKT phosphorylation, serum insulin etc.) are needed. These estimates will be particularly relevant in addressing the therapeutic potential of PAK4 inhibition to alleviate hepatic dysfunction from lipid accumulation.
4. To determine the impact of endogenous ketone bodies on PAK4 suppression, the authors have incubated primary hepatocytes with octanoate. Because octanoate is transported into the mitochondria in a CPT1 independent manner, will the results from these studies be the

same if the incubations are done either with a long chain fatty acid like palmitate or with beta-hydroxybutyrate?

5. Did the plasma ketone levels follow the same trend during feeding and fasting in all the three mice models of insulin resistance reported - HFD fed-, ob/ob and db/db-mice?

Response to Reviewers' Comments

Reviewer #1 (Remarks to the Author):

In this manuscript, Shi and colleagues investigated the role of PAK4-NCOR1/PPAR α signaling pathway in regulating ketogenesis during liver diseases. The authors found that PAK4 was increased in liver tissues of high fat diet-fed mice, NAFLD patients, and hepatocellular carcinoma patients. PAK4 phosphorylated NCoR1 on T1619/T2124 and induced its translocation towards nucleus to suppressed PPAR α and ketogenesis. PAK4 protein levels were significantly suppressed by fasting through either cAMP/PKA- or Sirt1-mediated ubiquitination and proteasome degradation. They provided evidence for a PAK4-NCOR1/PPAR α signaling pathway regulating ketogenesis. However, this research lacks enough innovation, and the conclusion of some aspects lacks sufficient evidence.

Main Points:

1. In Figure 1, the authors found levels of PAK4 protein in liver tissues/cells were regulated by fatty acid metabolism via ubiquitin-mediated degradation of PAK4, but not via transcription. Here, the research is too superficial, the key point is to identify the exact ubiquitinated sites in PAK4.

Response: In accordance to the reviewer's comments, we performed the additional experiments to identify lysine residues by which PKA activation mediates proteasomal degradation of PAK4. We substituted all four lysine residues in PAK4 with alanine and found that transfecting cells with either of K31A, K540A or K546A, but not K51A, completely abolished PKA mediated ubiquitination and degradation of PAK4 (Supplementary Figure 2a, b). These results suggest that K31, K540, and K546 of PAK4 are the responsible sites for ubiquitination and proteasome degradation of PAK4. We described this point in the Results section as follows;

To further identify the specific sites of ubiquitination in PAK4, we substituted four known lysine residues with alanine. The results revealed that K31, K540, and K546 of PAK4 are involved in the ubiquitination and subsequent proteasomal degradation (Supplementary Fig. 2a, b).

Additionally, the authors found p-S/T was associated PAK4 protein levels, what is the sites of p-S/T? Is it PAK4 S474 or the other sites? Is proteasome-dependent degradation of PAK4 resulted from the Ser/Thr phosphorylation of PAK4?

Response: We recently discovered that PAK4 in adipocytes was phosphorylated by PKA at two specific serine sites: S181 and S258 (Yu et al., Nature Metabolism, #NATMETAB-A22117822-R2). To investigate further, we generated the SA mutants and introduced them into AML12 cells. The findings demonstrated that only S258A mutant prevented the ubiquitination and degradation of PAK4 triggered by forskolin, while S181A and S474A mutants did not affect degradation (Figure 1h, i). These results suggest that PAK4 phosphorylation on S258 mediated by PKA is prerequisite for its ubiquitination and proteasome degradation. We collectively addressed these additional findings in the Results and discussion sections as follows;

In the Results section,

To determine the direct causality of Ser/Thr phosphorylation on PAK4 in protein degradation, we mutated candidate serine sites with alanine. The results indicated that S258 of PAK4 is the critical site responsible for PKA activation-induced ubiquitination and degradation of PAK4 (Fig. 1h, i).

In the Discussion section,

We found that PAK4 undergoes ubiquitin-proteasome degradation, with three lysine residues (K31, K540, or K546) playing a critical role, and MDM2 being closely involved. This degradation occurs following phosphorylation at S258 by PKA activated by glucagon. Moreover, it has been demonstrated that Sirt1 also mediates the ubiquitin-proteasome degradation of PAK4. Specifically, the treatment of hepatocytes with ketogenic substrates (such as octanoate and palmitate) or β OHB itself enhanced the expression of Sirt1, resulting in the deacetylation and degradation of PAK4.

2. PAK4 phosphorylates NCoR1 on T1619/T2124 according to the predictive tools, I think the authors should provide more rigorous approach such as IP-MASS to identify the phosphorylated sites in NCoR1.

Response: In response to the reviewer's comment, we performed an LC-MS/MS analysis and confirmed the phosphorylation on T1619 and T2124 residues of NCoR1 by PAK4 (Supplementary Figure 6a, Supplementary Table 5).

S6a

Supplementary Table 5. Identification of phosphorylation sites in NCoR1 by LC-MS/MS

Sequence	Charge state	Monoisotopic mass
QT(p)ILNDYITSQQMQVNL R	$[M+2H]^{2+}$	1123.0378
SPESQAQT(p)VLHPRPGSR	$[M+3H]^{3+}$	642.9776

3. β OHB acts as an endogenous inhibitor of histone deacetylases (HDACs) (PMID: 24140022). OCA results in increase of β OHB, which could inhibit activity of HDACs. Why the authors employed panobinostat (HDACs inhibitor) to rescue PAK4 decrease upon OCA treatment?

Response: As the reviewer pointed out, β OHB is generally regarded as an HDAC inhibitor. However, a class III HDAC Sirt1 has been reported to be upregulated by β OHB (Ref #21, Scheibye-Knudsen, M. et al. Cell Metab 2014) suggesting that the regulation of HDACs by β OHB is complicated depending on the cellular and environmental context. Indeed, our study indicated that β OHB's suppression of PAK4 was accompanied by an increase in Sirt1 expression, and was prevented by panobinostat (inhibitor of HDACs including Sirt1) but not by pracinostat (inhibitor of HDACs with no effect on Sirt1). This supports the notion that Sirt1 mediated β OHB's suppression of PAK4. To clarify this point, we revised the manuscript as follows;

β OHB has been shown to regulate various cellular functions as an endogenous modulator of histone deacetylase (HDAC), inhibiting class I/II/IV HDACs²⁰ or activating class III HDAC Sirt1²¹. We tested the involvement of HDAC in regulating PAK4 protein stability. Results showed that the repression of PAK4 induced by octanoate was abolished by panobinostat (a pan-inhibitor of the HDAC family)²², but not by pracinostat (a class I/II/IV HDAC inhibitor) (Supplementary Fig. 1j), indicating a role for class III HDAC sirtuins in PAK4 downregulation.

S1j

Ref #21. Scheibye-Knudsen, M. et al. A high-fat diet and NAD⁺ activate Sirt1 to rescue premature aging in cockayne syndrome. *Cell Metab* 20, 840-855, doi:10.1016/j.cmet.2014.10.005 (2014).

4. The author claimed that sirt1 is a regulator of PAK4, however, in fig.S1J sirt3 and sirt7 were unregulated and sirt4 is downregulated upon OCA treatment. Are there sirt3 and sirt7 paly role in PAK4 stabilization?

Response: We repeatedly measured the expression of Sirt3, Sirt4, and Sirt7 in hepatocytes after octanoate treatment and found that they are not significantly altered by octanoate, as shown in (a) of Figures for reviewer only. In our siRNA transfection study, we observed that unlike Sirt1 silencing, neither Sirt3 silencing nor Sirt7 silencing rescued octanoate-mediated PAK4 degradation (b). These results strongly indicate and support our original claim that Sirt1 is the sole isotype involved in PAK4 degradation upon octanoate exposure.

Figures for Reviewer only

a

b

In addition, cAMP/PKA/SIRT1 axis has been well elucidated (PMID: 33045622, PMID: 33369003). Why the author claimed cAMP/PKA and SIRT1 could play independently in regulating stabilization of PAK4.

Response: Supplementary Figure 1o demonstrated that glucagon treatment caused the phosphorylation and activation of Sirt1 (FoxO1 deacetylation), which was nullified by either PKA inhibitor H89 or insulin. These findings suggest the involvement of the cAMP-PKA-Sirt1 axis in the degradation of PAK4, as noted by the reviewer. However, interestingly, treatment with Sirt1 inhibitor EX-527 did not reverse the phosphorylation and suppression of PAK4 protein (Supplementary Figure 1p), ruling out the involvement of Sirt1 in glucagon's suppression of PAK4. Thus, we suggest that glucagon-activated cAMP-PKA pathway and β OHB-mediated Sirt1 upregulation independently play roles in the degradation of PAK4 (Figure 1k). We described this point in the Results section as follows;

Considering the apparent link between the cAMP-PKA pathway and Sirt1 activation^{23,24}, we further investigated the regulatory connections between these two pathways in PAK4 degradation. Glucagon treatment induced the Ser/Thr phosphorylation and activation of Sirt1 (deacetylation of FoxO1), which were also counteracted by either H89 or insulin (Supplementary Fig. 1o), implying the involvement of the cAMP-PKA-Sirt1 axis in PAK4 degradation. However, treatment with the Sirt1 inhibitor EX-527 did not impact the degradation of PAK4 by glucagon (Supplementary Fig. 1p), thus ruling out the possibility of Sirt1 activation being responsible for glucagon-mediated PAK4 repression. These results suggest that glucagon-mediated cAMP-PKA pathway and β OHB-mediated Sirt1 upregulation independently play roles in the ubiquitination/proteasome-mediated degradation of PAK4 (Fig. 1k).

5. In Fig.4f, why not show PAK4 S474A group using IF.

Response: As commented, we conducted an experiment where we introduced a PAK4 mutant (AdPAK4^{S474A}) into primary hepatocytes. Confocal microscopic findings revealed that PAK4 overexpression increased NCoR1 levels in the nucleus (Figure 4f). However, the overexpression of PAK4^{S474A} had no impact on this process, indicating that PAK4's effect on nuclear NCoR1 level is reliant on its kinase activity.

Additionally, the authors should test LaminB and GAPDH in all samples to confirm the quality of plasma and nucleus separation experiment.

Response: As suggested, we have now added Lamin B and GAPDH to cytosolic and nuclear extract samples respectively in Figure 4g, 5e, and 5g to verify the quality of nucleus/cytoplasm fractionation.

4g

5e

5g

6. In Fig. 5b, NcoR1 purified protein was not shown in CBB figure.

Response: We repeated the *in vitro* kinase assay and replaced the original figure with a new one. The new Coomassie blue staining clearly shows the presence of NcoR1 purified protein.

In Fig. 5F, in double mutant group, the total NCoR1 were decrease, which cannot conduct the decrease of NCoR1 nuclear translocation was induced by mutation of T1619A/T2124A. Besides, NCoR1 decreased significantly in fast and LKO group. Were NCoR1 were degraded through ubiquitin proteasome system under these conditions?

Response: We have added the quantification results of NCoR1 in whole cell lysates and cytosolic extracts to Figure 5e. As the reviewer pointed out, the total level of NCoR1 was lower in cells transfected with T1619A/T2124A compared to the wild-type NCoR1-transfected group. Furthermore, the PAK4-mediated increases and decreases of NCoR1 in the nucleus and cytosol, respectively, were also abolished in T1619A/T2124A. These results suggest that PAK4-mediated phosphorylation of NCoR1 on T1619/T2124 leads to its nuclear localization, preventing ubiquitination and subsequent proteasome degradation. To support this notion, both fasting and PAK4 ablation repressed the total level of NCoR1 in the liver, which was associated with increases in the ubiquitination and degradation of NCoR1, as shown in Figure 5g. We described this point in the Results section as follows;

Interestingly, we also noted a reduction in the total level of NCoR1 in cells transfected with the double-mutant. In addition, NCoR1 ubiquitination was dramatically increased in cells overexpressing double-mutant (Fig. 5f). **In mouse livers, we observed similar trends as observed in cells, where NCoR1 ubiquitination slightly increased during fasting but markedly increased in LKO mice (Fig. 5g). As a result, under fasting conditions, the phosphorylation and nuclear localization of NCoR1 were suppressed compared to the fed condition in WT mouse livers. Additionally, *Pak4* LKO mouse livers exhibited decreased phosphorylation, as well as reduced nuclear and total levels of NCoR1, even under fed conditions.** As the subcellular localization of NCoR1 is linked with its phosphorylation status, and several kinases have been implicated, we examined these kinase signaling pathways. Importantly, we found that the dephosphorylation of NCoR1 in *Pak4* LKO is not associated with the reduced activation of either mTORC1/S6 kinase 2 (S6K2) ^{6,9} or Akt ⁷, which are known kinases for this corepressor (Fig. 5g). These results demonstrate that PAK4 directly phosphorylates NCoR1 on T1619/T2124 and enhances its gene-repressive function by preventing its nuclear export **and the subsequent ubiquitination and proteasomal degradation.**

5e

5f

5g

7. The data of PAK4 roles in liver cancer are not solid. In Supplementary Fig. 7C, the authors should provide the picture of tumors.

Response: As commented, we have provided the picture of tumors in mice in Supplementary Figure 9d.

S9d

In addition, the xenograft tumor model used here is not enough. The authors should use spontaneous model of liver cancer with KD or NCD diet to explore the function of PAK4. Is NCoR1 T1619/T2124 involved in this process?

Response: We conducted an additional experiment to validate the roles of PAK4 and ketone bodies in liver cancer. In the orthotopic HCC model, we injected Hepa 1-6 cells directly into the livers of two groups of mice (WT and *Pak4* LKO) that were fed a normal chow diet (NCD). After one week, the mice were either maintained on an NCD or switched to a ketogenic diet (KD) for an additional two weeks (Supplementary Figure 9j). At the end of the experiment, we terminated the mice and compared the sizes of their tumors. Our results demonstrated that, similar to the extrahepatic tumor model (Supplementary Figure 9a-i), tumor size was significantly reduced, and β OHB levels were markedly increased with KD feeding

(Supplementary Figure 9k-q). These effects were even more pronounced in the *Pak4* knockout mice. We collectively described this point in the Results and Discussion sections as follows; In the Results section,

In an orthotopic tumor model, Hepa1-6 cells were directly injected into the livers of WT and *Pak4* LKO mice (Supplementary Fig. 9j). Similar to the subcutaneous tumor model, mice fed with a KD exhibited reduced tumor formation with higher β OHB levels (Supplementary Fig. 9k-q). These effects were even more pronounced in *Pak4* LKO mice. These findings suggest that circulating or intrahepatic ketone bodies regulated by PAK4 play a crucial inhibitory role in tumor growth.

Supplementary Figure 9. Suppression of extrahepatic and intrahepatic tumor growth in *Pak4* LKO mice. --- a, j. Schematic of the extra- (a) and intra-hepatic (j) tumor implantation models using *Pak4* LKO and WT mice fed a normal chow diet (NCD) or ketogenic diet (KD). --- k-m. Body weight change (k, $n = 4-5$), gross images of tumors (l, $n = 4-5$) and tumor weights (m, $n = 4-5$) at the end of the study. n, o. blood β OHB levels (n, $n = 4$) and its correlation with tumor weight (o, $n = 14$). p, q. Hepatic β OHB levels (p, $n = 4-5$) and its correlation with tumor weight (q, $n = 13$).

In the Discussion section,

There are a few limitations to this study. --- Secondly, only extrahepatic and intrahepatic tumor transplantation models were used in this study to investigate the role of PAK4 in regulating ketogenesis and its impact on tumor growth. Considering the limitations of traditional xenograft models for studying liver cancer, it may be necessary to employ spontaneous liver cancer models or humanized mouse models to establish a more robust translational implication of ketone bodies in tumor growth.

Minor points

1. The data is not rigorous and lacks control/input groups in some figures. For example, the input group is missed in Fig.S1K.

Response: As suggested, we have added input in all IP samples throughout the manuscript.

2. GAPDH is an enzyme involved in glycolysis. Glucagon and insulin will alter the expression and activity of this enzyme. Therefore, it is not appropriate to use GAPDH as an internal parameter.

Response: We replaced GAPDH with HSP90 in whole cell lysates in all the figures, except those involving HCC patients (Figure 7c and Supplementary Figure 10c). As shown below, HSP90 protein expression exhibited variability across individuals.

3. Lots of gene enriched in lipid metabolism were influenced in PAK4 LKO group, which may involve in regulating NAFLD pathology. Why the author put attention on ketogenesis and NCoR1?

Response: The process of ketogenesis in the liver can get rid of up to two-thirds of the lipids that enter it, and if it is not working properly, it can contribute to the development of non-alcoholic fatty liver disease (NAFLD). Studies suggest that ketogenesis in the liver can help reduce the severity of simple steatosis and NAFLD progression, but as NAFLD gets worse, the ability of the liver to produce ketone bodies declines (Mooli and Ramakrishnan). Therefore, understanding the mechanisms of hepatic ketogenesis is important for discovering new therapies for NAFLD. We showed that PAK4 inhibits ketogenesis, which leads to increased fat accumulation in liver tissue (Figure 2, Supplementary Figure 3). However, blocking PAK4 either genetically or pharmacologically can alleviate NAFLD in mice (Figure 6, Supplementary Figure 8, and Supplementary Figure 9). These findings demonstrate that impaired hepatic ketogenesis mediated by PAK4 plays a causal role in the development of NAFLD. We already described this point in the Introduction section.

Reference: Mooli RGR and Ramakrishnan SK Emerging role of hepatic ketogenesis in fatty liver disease. *Front Physiol.* 13:946474, 2022

4. The authors should analyze the nuclear localization of NCoR1 in high or low expression of PAK4 in human tissues by IF directly.

Response: We performed IF staining in human tissues and found that nuclear level of NCoR1 was higher in case with high expression of PAK4 (Supplementary Figure 10d).

S10d

5. In GSEA, $FDR < 0.25$ was considered a trusted enrichment. Fig.3a, the FDR was 0.489 far from 0.25 in GSEA about Regulation of Ketone Biosynthetic Process. So not sure if it is relevant to Ketogenesis. It is best to display heat map of genes that enriched in this pathway and detect their expression.

Response: As commented, we have provided heatmap of GSEA of DEGs based on RNAseq data in response to 24-hour fasting in Supplementary Figure 4d.

S4d

Reviewer #2 (Remarks to the Author):

In their manuscript titled “p21-activated kinase 4 suppresses ketogenesis by phosphorylating NCoR1”, She and co-authors build a compelling story demonstrating that PAK4 kinase, previously described as an oncoprotein, phosphorylates nuclear corepressor NCoR1 leading to an increase of its nuclear localization that leads to inhibition of ketogenesis through transcriptional repression of PPAR alpha.

The authors demonstrate that levels of hepatic PAK4 are increased in mouse models of obesity and fatty liver, such as HFD-induced and genetically induced (Ob/ob and db/db) obesity. PAK4 protein is downregulated in fasting through two independent mechanisms (PKA and Sirt1) thus allowing appropriate physiological activation of ketogenesis during fasting.

Overexpression of PAK4 (but not the kinase-inactive mutant) leads to decreased expression of enzymes in fatty acid oxidation and ketogenic pathways and fat accumulation in the liver, particularly in conditions where ketogenesis is normally activated, such as fasting and low carb (ketogenic) diet feeding.

Liver-specific KO of PAK4 leads to enhanced expression of ketogenic enzymes in the liver, increased serum B-OHB and decreased hepatic and circulating TGs during fasting and on ketogenic diet. RNAseq analysis identified PPAR alpha as the key factor connecting the pathways affected by PAK4 KO. PAK4, but not the kinase-inactive mutant suppressed PPAR alpha in luciferase assays, but did not affect phosphorylation of PPAR alpha itself. Proximity ligation assays and Co-Ips identified nuclear receptor corepressor NCoR1 as the phosphorylation target of PAK4. This event increases nuclear localization of NCoR1 and facilitates its interactions with PPAR alpha leading to repression of its transcriptional activity. Increased binding of NCoR1 to PPAR alpha response elements by overexpression of PAK4 was also confirmed through ChIP.

Effects of PAK4 overexpression on ketogenesis were blunted by knock-down or overexpression of NCoR with mutated PAK4 phosphorylation sites.

The authors also noted decreased TG accumulation and increased circulating BOHB in the livers of mice with hepatic KO of PAK4 fed with HFD.

A small molecule inhibitor of PAK4 recapitulated effects of hepatic PAK4 KO on ketogenesis in mice fed ketogenic and high fat obesogenic diets.

Importantly, authors also demonstrate that increased ketogenesis through KO of hepatic PAK4 decreases extrahepatic tumor growth in mice fed ketogenic diets.

WB analysis of biopsies material from human HCC demonstrated higher expression of PAK4 and NCoR1 and lower levels of HMGCS2 in tumor compared to non-tumor tissues. In line with these and previous finding in human HCC population, overall and relapse-free survival were significantly poorer in HCC patients with high expression levels of PAK4.

This is a well-written study with exciting new findings and potentially important ramifications for human health. The study is well designed, executed, and presented. The conclusions are well-supported by the data. The methodology is sound, and methods are described in sufficient detail. Limitations of the study are also acknowledged by the authors.

Response: We thank the reviewer for the positive comments and suggestions.

Below are some more general comments and suggestions.

1. The basic physiologic model proposed by the authors describes an elegant mechanism for additional control of beta-oxidation and ketogenesis, where PAK4 is abundant and active in fed conditions. It phosphorylates NCoR1 promoting its interactions with PPAR alpha to suppress ketogenesis in fed animals, where it is physiologically irrelevant. During fasting PAK4 levels drop allowing for de-phosphorylation of NCoR and its dissociation from PPAR alpha, so ketogenic gene expression program can be activated. However, the strongest phenotypic effects of hepatic PAK4 KO on liver and serum TGs, and circulating B-OHB are observed in conditions where ketogenesis is already active: in fasting and on ketogenic diet. mRNA and protein expression of relevant enzymes are also shown under these conditions. Can the authors elaborate on this apparent discrepancy between the basic model and the conditions where experimental observations were made?

Response: In this study, we demonstrated that inhibition of PAK4, either genetically (Figure 3c and Supplementary Figure 5d) or pharmacologically (Supplementary Figure 8g), slightly but significantly increased blood β OHB levels compared to their control mice even in the fed state. During fed states, animals preferentially burn carbohydrate to generate ATP, and fatty acids (either dietary-derived or those converted from surplus carbohydrate) are stored as triacylglycerides in adipose tissue. Moreover, fatty acid β -oxidation is largely repressed under fed conditions through inhibition of carnitine palmitoyltransferase 1 (CPT1) by malonyl-CoA, an intermediate of fatty acid biosynthesis. Therefore, the effect of PAK4 inhibition for ketogenesis is minimal in fed conditions. It is noteworthy that PPAR α KO mice have shown a profound impairment of fatty acid oxidation and ketogenesis only when fasted state (Ref # 3), consistent with our results of PAK4 overexpression (Figure 2a). However, in response to fasting and ketogenic diet feeding conditions in which lipid substrates for β -oxidation and ketogenesis become abundant, this difference was amplified to a greater extent. These findings suggest that the suppression of ketogenesis in a fed state is attributed to the abundant PAK4 level at least partially, and this inhibition is released during fasting or when ketogenesis is initiated. We already described this point in the Discussion section as follows;

PAK4 puts a brake on hepatic ketogenesis under fed conditions and that this brake is released upon fasting or starting ketogenesis, in the following sequence of events: fasting \rightarrow glucagon/PKA activation and β OHB/Sirt1 induction \rightarrow PAK4 degradation \rightarrow NCoR1 de-repression \rightarrow PPAR α transactivation \rightarrow β OHB production.

Reference #3

Hashimoto, T. et al. Defect in peroxisome proliferator-activated receptor alpha-inducible fatty acid oxidation determines the severity of hepatic steatosis in response to fasting. *J Biol Chem* 275, 28918-28928, doi:10.1074/jbc.M910350199 (2000).

2. Along the lines of the first point, it would be interesting to hear if the authors could comment on whether PAK4-mediated phosphorylation of NCoR1 also affects interactions of this

corepressor with other nuclear receptors, whose functions are reciprocal to PPAR alpha (LXR, THR)?

Response: In this question, the reviewer asked if PAK4 mediated phosphorylation of NCoR1 specifically affects PPAR α target genes or affects other nuclear receptors interacting with NCoR1 such as LXR α and THR β . Co-IP experiments conducted in primary hepatocytes revealed that the overexpression of PAK4 had no effect on the interaction between NCoR1 and LXR α , while it led to a decrease in the interaction between NCoR1 and THR β (Figure 4e). These findings suggest a potential broad impact of PAK4 regulation on NCoR1 in various pathogenic processes. We described this point in the Results section as follows;

On the contrary, the binding of NCoR1 to the other nuclear receptor, thyroid hormone receptor beta (THR β), was decreased by PAK4 overexpression, showing a reciprocal regulation compared to PPAR α . The interactions between NCoR1 with LXR α , PPAR α with p300, or PPAR α with SMRT remained unchanged (Fig. 4d, e). Considering that the NCoR1-THR β pathway is crucial for hepatic lipid synthesis and storage^{30,31}, further study is needed to understand the regulation of PAK4 on this pathway.

3. Finally, the authors should be cautious when using term “nuclear translocation” describing effects of PAK4 on NCoR1, as there is no data in this manuscript that addresses the exact mechanism of the changes of intracellular localization of NCoR1.

Response: We appreciate the Reviewer’s meticulous reading of our manuscript and used the term “nuclear level” instead of “nuclear translocation”.

Reviewer #3 (Remarks to the Author):

In this manuscript, the authors detail a mechanism through which p21 activated kinase 4 (PAK4) regulates ketogenesis through the interaction between nuclear receptor corepressor 1 (NCoR1) and PPAR α . The manuscript details elegant studies to identify the above interactions and the phosphorylation sites of NCoR1 leading to its impact on ketogenesis. While the detailed molecular mechanisms are elaborate and robust, the authors should address the following concerns.

All the metabolic impacts detailed in the manuscript clearly point to lipid oxidation (β -oxidation) rather than ketogenesis as the major driver of the observed effects in the liver. While the majority of ketogenesis results from break down of fatty acids, the mechanisms proposed in the manuscript seem to modulate the pathway of β -oxidation rather than ketogenesis. The authors should clarify the rationale for focusing on ketogenesis rather than lipid oxidation. If the proposed mechanisms have a specific impact on ketogenesis, wouldn't it be also logical to explore the PPAR α mediated regulation of FGF-21 in these experiments? The authors should clearly delineate the impacts of the PAK4-NCoR1/PPAR α axis on β -oxidation vs. ketogenesis

Response: We thank the reviewer for this critical comment. In accordance with the reviewer's comment, we assessed the serum FGF21 levels in mice overexpressed with PAK4, as well as *Pak4* LKO mice. Our findings indicate that PAK4 overexpression repressed serum levels of FGF21 (Figure 2f). On the contrary, *Pak4* LKO fed mice displayed the elevated levels of FGF21 compared to those of WT mice, while fasting or KD feeding equally increased FGF21 with no changes between genotypes (Figure 3g). Consistent with the results of PPAR α -luciferase assay (Supplementary Figure 8d), the overexpression of PAK4 significantly decreased FGF21-luciferase activity (Figure 2i). We also assessed whether the effect of the PAK4 inhibitor on ketogenesis is altered in PPAR α KO mice and found that the induction of ketogenesis by the PAK4 inhibitor was completely compromised in PPAR α KO mice (Supplementary Figure 8g). Furthermore, the silencing of FGF21 did not compromise the increase in ketogenesis in *Pak4* LKO hepatocytes (Supplementary Figure 8j).

We changed the title to “p21-activated kinase 4 suppresses fatty acid oxidation and ketogenesis by phosphorylating NCoR1”, and revised the manuscript to reflect this notion.

In the Results section,

Given the crucial role of FGF21 in ketogenesis through a mechanism partially independent of PPAR α ³, along with the regulatory effect of PAK4 on FGF21 expression (Fig. 2f, i, 3g), we explored the potential involvement of FGF21 in the role of PAK4. Following ND201651 treatment, there was a notable increase in blood levels of β OHB, accompanied by elevated protein levels of CPT1 α and HMGCS2 in WT liver tissues (Supplementary Fig. 8g–i). However, these effects were nullified in *Ppara* KO mice. In addition, the increased production of ketone bodies in *Pak4* LKO hepatocytes remained unchanged even after silencing FGF21 (Supplementary Fig. 8j, k). These findings suggest that the inhibition of PAK4 specifically impacts the PPAR α -dependent fatty acid β -oxidation pathway, leading to the production of β OHB, while implying that FGF21 does not exert a significant influence on these processes. Altogether, these studies provide compelling evidence for the therapeutic potential of the PAK4 inhibitor in treating fatty liver conditions associated with impaired ketogenesis.

In the Discussion section,

We also discovered that PAK4 inhibition can attenuate hepatic fat accumulation, by enhancing fatty acid β -oxidation and ketogenesis, which correlates with reduced phosphorylation of

NCoR1. Of note, we present evidence that PAK4 inhibition-enhancement of fatty acid β -oxidation is the major driver of the ketogenesis and the consequent improvement of fatty liver. Firstly, the increase in β OHB synthesis induced by the PAK4 inhibitor was nullified in *Ppara* KO mice (Supplementary Fig. 8g). This contrasts with FGF21, which promotes ketogenesis partly through a PPAR α -independent mechanism⁴⁰. Secondly, silencing FGF21 did not alter the elevated β OHB production observed in *Pak4* LKO hepatocytes (Supplementary Fig. 8j). This suggests that, although *Pak4* LKO mice exhibited increased FGF21 levels under fed conditions mimicking fasting (Fig. 3g), likely through PPAR α activation, FGF21 did not play a critical role in mediating the ketogenesis induced by PAK4 inhibition.

Supplementary Figure 8. Metabolic phenotypes in C57BL/6 mice or *Ppara* KO mice administered with PAK4 inhibitor ND201651 (compare to Figure 6). ----- . **g-i.** *Ppara* KO mice and their littermates (WT) were orally administered ND201651 (ND, 50 mg/kg) once a day for 3 days. Blood levels of β OHB (g, $n = 3$) and glucose (h, $n = 3$), and protein levels of CPT1 α , HMGCS2 and PPAR α in liver tissues were analyzed after *ad libitum* feeding or 24-h fasting (i). **j, k.** *Pak4* LKO and WT hepatocytes were transfected with siRNA targeting FGF21 or control siRNA, and the medium β OHB concentration was assessed (j, $n = 6$). Successful depletion of FGF21 was validated by Western blotting (k).

2. While the broad idea of correcting defects in ketogenesis/lipid oxidation through the modulation of the PAK4-NCoR1/PPAR α axis is attractive, the proposed mechanisms will induce mitochondrial activity. The impacts of chronic induction of ketogenesis/lipid oxidation on

mitochondrial activity/OXPHOS, hepatocellular stress/ inflammation is not explored, especially in the context of evaluating the therapeutic potential of these mechanisms.

Response: In response to reviewer's comment, we examined the levels of OxPhos proteins, markers of ER stress and the inflammatory genes in the liver tissues of WT and *Pak4* LKO mice that were fed a ketogenic diet for two weeks. Results showed that ER stress/inflammatory genes were reduced in LKO compared to those in WT mice, although OxPhos proteins did not differ between genotypes. We described this point in the Results and Discussion sections as follows;

In the Results section,

In harmony with these findings, hepatic stress was reduced in *Pak4* LKO mice under KD conditions, as evidenced by the mRNA levels of inflammation-related genes and protein levels of endoplasmic reticulum stress genes (Supplementary Fig. 4k, l).

In the Discussion section,

Interestingly, mice that overexpress PAK4 exhibited increased hepatic TG levels during fasting or on a KD, phenocopying mice lacking PPAR α ³. In contrast, *Pak4* LKO mice or mice treated with a PAK4 inhibitor exhibited attenuated fat accumulation in the liver during fasting, on a KD, or HFD. Consistent with this, the levels of inflammatory genes and endoplasmic reticulum stress proteins were reduced in KD-fed *Pak4* LKO mice compared to WT mice. Our results collectively suggest that inhibiting PAK4 might be a therapeutic target for preventing hepatic steatosis.

Figure for reviewer only

3. The broad theme of the manuscript investigates the impact of the proposed mechanisms mediated by PAK4 on lipid accumulation in the liver. Towards determining clearly, the beneficial impact of the proposed mechanisms on lipid metabolic network in the liver, estimates of changes in insulin sensitivity/ signaling (e.g., AKT phosphorylation, serum insulin etc.) are needed. These estimates will be particularly relevant in addressing the therapeutic potential of PAK4 inhibition to alleviate hepatic dysfunction from lipid accumulation.

Response: In response to the reviewer's suggestion, we have included the mRNA levels of genes involved in *de novo* lipogenesis (ACC, FAS, LXR α , SCD1, and SREBP1c) that remained unchanged upon PAK4 overexpression or in *Pak4* LKO during fasting (Supplementary Figure 3d, 4m). These data indicate that the reduced lipid accumulation in the liver in *Pak4* LKO largely resulted from increased lipid oxidation rather than lipogenesis.

Further, we investigated the effect of *Pak4* deletion on insulin signaling in mice fed with a high-fat diet. Our findings indicate that, while HFD-fed WT mice exhibited reduced p-Akt levels, *Pak4* LKO mice demonstrated improved p-Akt levels without alterations in serum insulin levels (Supplementary Figure 7f-h). We described this point in the Results section as follows;

In HFD-fed LKO mice, the level of Akt phosphorylation in the liver was higher compared to HFD-fed WT mice (Supplementary Fig. 7f). There were no differences in serum insulin levels and the Homeostatic Model Assessment of Insulin Resistance (HOMA-IR) index between the genotypes (Supplementary Fig. 7g, h).

4. To determine the impact of endogenous ketone bodies on PAK4 suppression, the authors have incubated primary hepatocytes with octanoate. Because octanoate is transported into the mitochondria in a CPT1 independent manner, will the results from these studies be the same if the incubations are done either with a long chain fatty acid like palmitate or with beta-hydroxybutyrate?

Response: To address this inquiry, we cultured primary hepatocytes with palmitate and examined the level of β OHB in the medium. We noted a gradual decrease in PAK4 protein level and an increase in Sirt1 level by treatment with either palmitate or β OHB (Supplementary Figure 1l). Concurrently, we observed a significant increase in the β OHB level after a 6 or 12-hour incubation with palmitate, although to a lesser degree compared to octanoate (Supplementary Figure 1m). The contribution of Sirt1 in octanoate-mediated PAK4 repression was validated in a Sirt1 silencing study (Supplementary Figure 1n). We described this point in the Results section as follows:

Treatment with the long-chain fatty acid palmitate or β OHB also resulted in a reduction in PAK4 protein levels, showing an inverse correlation with Sirt1 protein levels (Supplementary Fig. 1l).

Palmitate treatment confirmed an increase in ketone body synthesis, although to a lesser degree compared to octanoate (Supplementary Fig. 1m). Importantly, the decrease in PAK4 protein levels induced by octanoate was reversed upon Sirt1 silencing (Supplementary Fig. 1n). Similar to the findings in the glucagon study, treatment with octanoate also led to PAK4 protein degradation via the ubiquitination pathway, and this process was dependent on Sirt1 activation (Fig. 1f, g).

5. Did the plasma ketone levels follow the same trend during feeding and fasting in all the three mice models of insulin resistance reported - HFD fed-, *ob/ob* and *db/db*-mice?
Response: As shown in Supplementary Figure 7a in the revised manuscript, fasting-induced blood ketone body levels in obese HFD-fed, *ob/ob*, and *db/db* mice were significantly lower compared to those in the corresponding control mice, which was associated with an upregulation of PAK4 protein levels in the liver. We described this point in the Results section as follows; In line with our findings of increased PAK4 expression in mice with fatty liver (HFD-fed, *ob/ob*, and *db/db* mice, Fig. 1c), we also observed defective ketogenesis in these mice (Supplementary Fig. 7a).

S7a

REVIEWERS' COMMENTS

Reviewer #1 (Remarks to the Author):

The authors have addressed all my concerns, I don't have any other questions.

Reviewer #2 (Remarks to the Author):

I thank the authors for their thoughtful responses and additional data related to my questions.

Reviewer #3 (Remarks to the Author):

The authors have done a commendable job in terms of addressing all my concerns and questions in the revised version of the manuscript. I have not further questions and concerns.